# UniFormerV2: Spatiotemporal Learning by Arming Image ViTs with Video UniFormer

## Abstract

Learning discriminative spatiotemporal representation is the key problem of video understanding. Recently, Vision Transformers (ViTs) have shown their power in learning long-term video dependency with self-attention. Unfortunately, they exhibit limitations in tackling local video redundancy, due to the blind global comparison among tokens. UniFormer has successfully alleviated this issue, by unifying convolution and self-attention as a relation aggregator in the transformer format. However, this model has to require a tiresome and complicated image-pretraining phrase, before being finetuned on videos. This blocks its wide usage in practice. On the contrary, open-sourced ViTs are readily available and well-pretrained with rich image supervision. Based on these observations, we propose a generic paradigm to build a powerful family of video networks, by arming the pretrained ViTs with efficient UniFormer designs. We call this family UniFormerV2, since it inherits the concise style of the UniFormer block. But it contains brand-new local and global relation aggregators, which allow for preferable accuracy-computation balance by seamlessly integrating advantages from both ViTs and UniFormer. Without any bells and whistles, our UniFormerV2 gets the state-of-the-art recognition performance on 8 popular video benchmarks, including scene-related Kinetics-400/600/700 and Moments in Time, temporal-related Something-Something V1/V2, untrimmed ActivityNet and HACS. In particular, it is the first model to achieve 90% top-1 accuracy on Kinetics-400, to our best knowledge. The models will be released afterward.

## 1 Introduction

Spatiotemporal representation learning is a fundamental task in video understanding. Recently, Vision Transformers (ViTs) have achieved remarkable successes in the image domain (Dosovitskiy et al., 2021; Wang et al., 2021b; Liu et al., 2021; Li et al., 2022a). Therefore, researchers make a great effort to transfer image-based ViTs for video modeling (Bertasius et al., 2021; Arnab et al., 2021; Yan et al., 2022), by extending Multi-Head Self-Attention (MHSA) along the temporal dimension. However, the spatiotemporal attention mechanism in these approaches mainly focuses on capturing global video dependency, while lacking the capacity of tackling local video redundancy. As a result, these models bear a large computational burden to encode local video representations in the shallow layers, leading to unsatisfactory accuracy-efficiency balance in spatiotemporal learning.

To tackle these problems, researchers introduce a concise UniFormer (Li et al., 2022a), which unifies convolution and self-attention as Multi-Head Relation Aggregator (MHRA) in a transformer fashion. By modeling local and global relations respectively in shallow and deep layers, it can not only learn discriminative spatiotemporal representation but also largely reduce computation burden. However, as a new architecture for video modeling, UniFormer does not have any image-based pretraining as a start. To obtain a robust visual representation, it has to go through a tedious supervised pretraining phase by learning images from scratch, before finetuning on videos. Alternatively, we notice that there are various open-sourced image ViTs (Wightman, 2019; Touvron et al., 2021), which have been well-pretrained on huge web datasets under rich supervision such as image-text contrastive learning (Radford et al., 2021) and mask image modeling (He et al., 2022; Bao et al., 2021). These models exhibit great generalization capacity on a range of vision tasks (Luo et al., 2022; Chen et al., 2022; Shen et al., 2021). Hence, we are motivated by a natural question: *Can we integrate advantages from both ViTs and UniFormer for video modeling?*

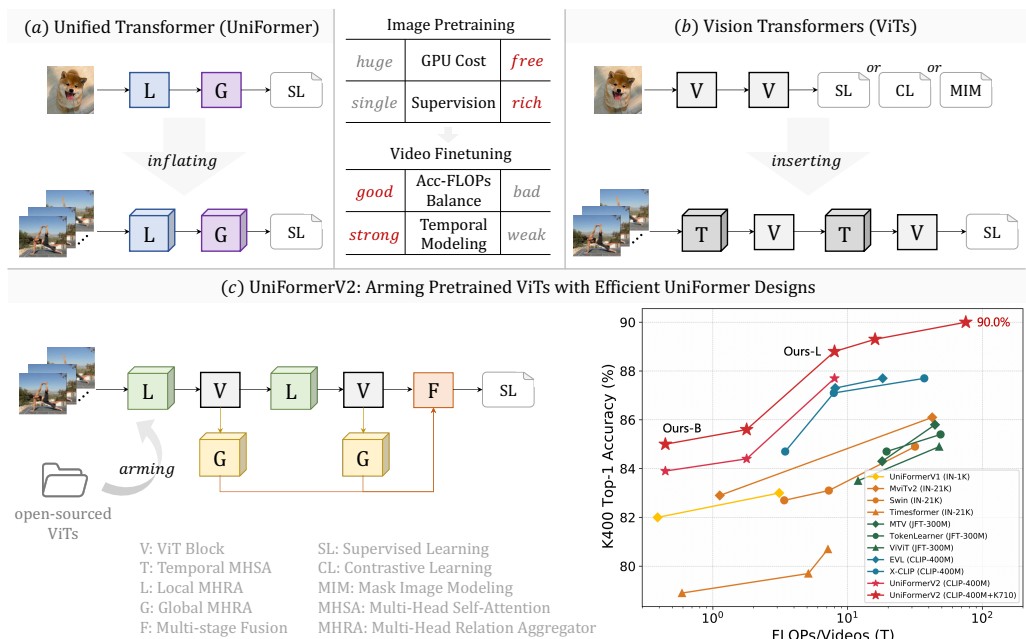

Figure 1: **Comparison on video modeling paradigm.** UniFormerV1 requires costly image pre-training, while directly inserting temporal MHSA into ViTs struggles for accuracy-FLOPs balance. UniFormerV2 can effectively and efficiently arm well-pretrained ViTs with concise UniFormer designs, thus integrating advantages from both models for spatiotemporal representation learning. To our best knowledge, it is the first model that achieves **90.0%** top-1 accuracy on Kinetics-400.

In this paper, we propose a generic paradigm to construct a powerful family of video networks, by arming the image-pretrained ViTs with efficient video designs of UniFormer. We called the resulting model UniFormerV2 (Fig. 1), since it inherits the concise style of UniFormer but equips local and global UniBlocks with new MHRA. In the local UniBlock, we flexibly insert a local temporal MHRA before the spatial ViT block. In this case, we can largely reduce temporal redundancy as well as leverage the well-pretrained ViT block, for learning local spatiotemporal representation effectively. In the global UniBlock, we introduce a query-based cross MHRA. Unlike the costly global MHRA in the original UniFormer, our cross MHRA can summarize all the spatiotemporal tokens into a video token, for learning global spatiotemporal representation efficiently. Finally, we re-organize local and global UniBlocks as a multi-stage fusion architecture. It can adaptively integrate multi-scale spatiotemporal representation to capture complex dynamics in videos.

We deploy our paradigm on ViTs that are pretrained on three popular supervision, including supervised learning, contrastive learning, and mask image modeling. All the enhanced models have great performance on video classification, showing the generic property of our UniFormerV2. Moreover, we develop a compact Kinetics-710 benchmark, where we integrate action categories of Kinetics-400/600/700, and remove the repeated and/or leaked videos in the training sets of these benchmarks for fairness (i.e., the total number of training videos is reduced from 1.14M to 0.66M). After training on K710, our model can simply achieve higher accuracy on K400/600/700 via only 5-epoch fine-tuning. Finally, extensive experiments show that, our UniFormerV2 achieves state-of-the-art performance on 8 popular video benchmarks, including scene-related datasets (i.e., Kinetics-400/600/700 (Carreira & Zisserman, 2017; Carreira et al., 2018; 2019) and Moments in Time (Monfort et al., 2020)), temporal-related datasets (i.e., Something-Something V1/V2 (Goyal et al., 2017b)), and untrimmed datasets (i.e., ActivityNet (Heilbron et al., 2015) and HACS (Zhao et al., 2019)). To our best knowledge, it is the first model to achieve **90.0%** top-1 accuracy on Kinetics-400.

## 2 RELATED WORK

**Vision Transformer.** Following Transformer in NLP (Vaswani et al., 2017), Vision Transformer (ViT) (Dosovitskiy et al., 2021) has made great successes in various vision tasks, including object detection Carion et al. (2020); Zhu et al. (2021), semantic segmentation Xie et al. (2021); Cheng

et al. (2021), low-level image processing Liang et al. (2021); Cui et al. (2022), action recognition (Bertasius et al., 2021; Arnab et al., 2021), temporal localization (Zhang et al., 2022) and multi-modality learning (Radford et al., 2021; Wang et al., 2022). To make ViT more efficient and effective, researchers introduce scale and locality modeling in different ways, such as multi-scale architectures (Wang et al., 2021b; Fan et al., 2021), local window (Liu et al., 2021), early convolution embedding (Xiao et al., 2021; Yuan et al., 2021a) and convolutional position encoding (Chu et al., 2021; Dong et al., 2022). Alternatively, UniFormer (Li et al., 2022a) unifies convolution and self-attention as relation aggregator in a transformer manner, thus reducing large local redundancy.

**Video Learning.** 3D Convolutional Neural Networks (CNNs) once played a dominant role in video understanding (Tran et al., 2015; Carreira & Zisserman, 2017). Due to the difficult optimization problem of 3D CNNs, great efforts have been made to factorize 3D convolution in the spatiotemporal dimension (Tran et al., 2018; Qiu et al., 2017; Feichtenhofer et al., 2019) or channel dimension (Tran et al., 2019; Feichtenhofer, 2020; Kondratyuk et al., 2021). However, the local receptive field limits 3D convolution to capture long-range dependency. The global attention motivates researchers to transfer image-pretrained ViTs to video tasks (Bertasius et al., 2021; Neimark et al., 2021; Zhang et al., 2021b; Arnab et al., 2021; Bulat et al., 2021; Patrick et al., 2021). To make the video transformer more efficient, prior works introduce hierarchical structure with pooling self-attention (Fan et al., 2021), local self-attention (Liu et al., 2022) or unified attention (Li et al., 2022a). Though these novel models are adept at temporal modeling, they rely on tiresome image pretraining. In contrast, various well-pretrained ViTs with rich supervision are open-sourced (Wightman, 2019). In this paper, we aim to extend efficient UniFormer designs to ViT, arming it as a strong video learner.

## 3 METHOD

**Overall Framework**. We propose to arm an image ViT with video designs of UniFormer (Li et al., 2022a), leading to UniFormerV2. On one hand, spatial interactions in well-pretrained ViT can be fully leveraged and preserved to enhance spatial modeling. On the other hand, hierarchical temporal interactions in efficient UniFormer can be flexibly adopted to enhance temporal modeling. Our overall architecture is shown in Fig. 2. It firstly projects input videos into tokens, then conducts local and global modeling by the corresponding UniBlocks. Finally, a multi-stage fusion block will adaptively integrate global tokens of different stages to further enhance video representation.

Specifically, we first use 3D convolution (i.e., $3 \times 16 \times 16$) to project the input video as $L$ spatiotemporal tokens $\mathbf{X}^{in} \in \mathbb{R}^{L \times C}$, where $L = T \times H \times W$ ($T$, $H$, and $W$ respectively denote temporal, height, and width). Following the original ViT design (Dosovitskiy et al., 2021), we perform spatial downsampling by a factor of 16. For better temporal modeling, we conduct temporal downsampling by a factor of 2. Next, we construct the local and global UniBlocks. For our local block, we reformulate the image-pretrained ViT block, by inserting the local temporal MHRA (Li et al., 2022a) before it. In this case, we can effectively leverage the robust spatial representation of ViT as well as efficiently reduce local temporal redundancy. Moreover, we introduce a global UniBlock on top of each local UniBlock, which can capture full spatiotemporal dependency. For computational efficiency, we design a query-based cross MHRA to aggregate all the spatiotemporal tokens as a global video token. All these tokens with different-level global semantics from multiple stages are further fused for discriminative video representation.

### 3.1 LOCAL UNIBLOCK

To efficiently model temporal dependency upon the well-learned spatial representation, we propose a new local UniBlock, by inserting a local temporal MHRA before the standard ViT block,

$$\mathbf{X}^T = \text{LT\_MHRA}\left(\text{Norm}\left(\mathbf{X}^{in}\right)\right) + \mathbf{X}^{in}, \tag{1}$$

$$\mathbf{X}^S = \text{GS\_MHRA}\left(\text{Norm}\left(\mathbf{X}^T\right)\right) + \mathbf{X}^T, \tag{2}$$

$$\mathbf{X}^L = \text{FFN}\left(\text{Norm}\left(\mathbf{X}^S\right)\right) + \mathbf{X}^S. \tag{3}$$

LT\_MHRA and GS\_MHRA refer to MHRA with local temporal affinity and global spatial affinity respectively. FFN consists of two linear projections separated by GeLU (Hendrycks & Gimpel, 2016). Additionally, following the normalization in UniFormer (Li et al., 2022a), we adopt Batch Norm (BN) (Ioffe & Szegedy, 2015) before local MHRA, and Layer Norm (LN) (Ba et al., 2016)

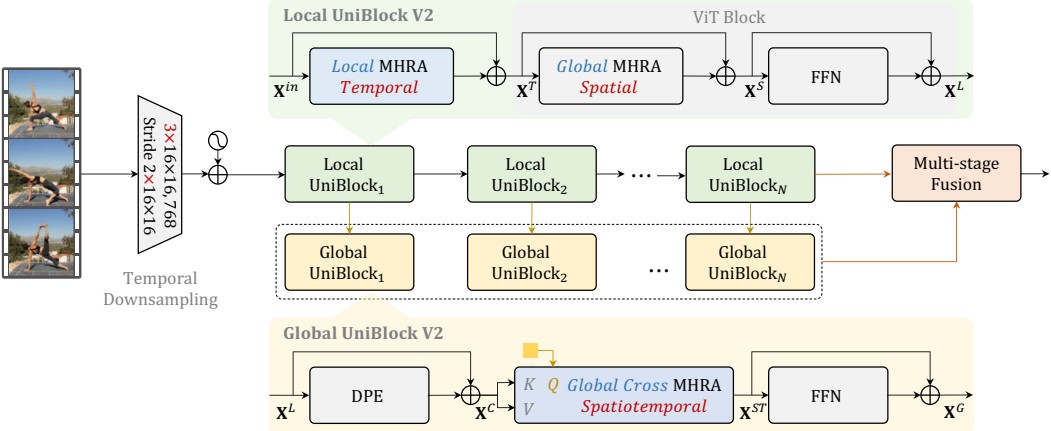

Figure 2: **Overall framework of our UniFormerV2.** There are three key blocks, i.e., local and global UniBlocks, and multi-stage fusion block. All these designs are efficient and effective.

before global MHRA and FFN. Note that GS_MHRA and FFN come from the image-pretrained ViT block. In general, MHRA (Li et al., 2022a) learn token relation via multi-head fusion:

$$R_n(\mathbf{X}) = A_n V_n(\mathbf{X}), \tag{4}$$

$$\text{MHRA}(\mathbf{X}) = \text{Concat}(R_1(\mathbf{X}); R_2(\mathbf{X}); \cdots; R_N(\mathbf{X}))\mathbf{U}, \tag{5}$$

where $R_n(\cdot)$ refers to the relation aggregator in the $n$-th head. $A_n$ is an affinity matrix that describes token relation and $V_n(\cdot)$ is a linear projection, while $\mathbf{U} \in \mathbb{R}^{C \times C}$ is a learnable fusion matrix. For our local UniBlock, we insert LT_MHRA to reduce local temporal redundancy, which shares a similar design insight with the original UniFormer (Li et al., 2022a). Hence, the affinity in LT_MHRA is local with a learnable parameter matrix $a_n \in \mathbb{R}^{t \times 1 \times 1}$ in the temporal tube $t \times 1 \times 1$,

$$A_n^{\text{LT}}(\mathbf{X}_i, \mathbf{X}_j) = a_n^{i-j}, \ \ where \ \ j \in \Omega_i^{t \times 1 \times 1}. \tag{6}$$

This allows to efficiently learn the local temporal relation between one token $\mathbf{X}_i$ and other tokens $\mathbf{X}_j$ in the tube. Alternatively, GS_MHRA belongs to the original ViT block. Therefore, the affinity in GS_MHRA refers to a global spatial self-attention in the single frame $1 \times H \times W$,

$$A_n^{\text{GS}}(\mathbf{X}_i, \mathbf{X}_j) = \frac{\exp\{Q_n(\mathbf{X}_i)^T K_n(\mathbf{X}_j)\}}{\sum_{j' \in \Omega_{1 \times H \times W}} \exp\{Q_n(\mathbf{X}_i)^T K_n(\mathbf{X}_{j'})\}}, \tag{7}$$

where $Q_n(\cdot)$ and $K_n(\cdot) \in \mathbb{R}^{L \times \frac{C}{N}}$ are different linear projections in the $n$-th head.

**Discussion**. **(I)** Note the spatiotemporal affinity in our local UniBlock is decomposed as local temporal one $A_n^{\text{LT}}$ in Eq. (6), and global spatial one $A_n^{\text{GS}}$ in Eq. (7). In this case, we can not only leverage the efficient video processing design of UniFormer but also inherit the effective image pre-training of ViT. Alternatively, such local affinity in the original UniFormer (Li et al., 2022a) is jointly spatiotemporal, i.e., $A_n^{local}(\mathbf{X}_i, \mathbf{X}_j) = a_n^{i-j}$, where $j$ belongs to a 3D tube $\Omega_i^{t \times h \times w}$. The parameter matrix has to learn from scratch, which inevitably increases the training cost. **(II)** Compared with UniFormer, we abandon its Dynamic Position Encoding (DPE) in the local UniBlock, since the position encoding in the ViT block has characterized token locations. Table 9b also reveals an extra DPE in the local UniBlock does not help. **(III)** Instead of applying global temporal modeling as in TimeSformer (Bertasius et al., 2021), we use local affinity for temporal characterization, largely reducing the computation burden by tackling temporal redundancy in the UniFormer style.

## 3.2 GLOBAL UNIBLOCK

To explicitly conduct long-range dependency modeling on the spatiotemporal scale, we introduce a global UniBlock in our UniFormerV2. Specifically, this global UniBlock consists of three basic components including DPE, MHRA, and FFN as follows,

$$\mathbf{X}^C = \text{DPE}\left(\mathbf{X}^L\right) + \mathbf{X}^L, \tag{8}$$

$$\mathbf{X}^{ST} = \text{C\_MHRA}\left(\text{Norm}\left(\mathbf{q}\right), \text{Norm}\left(\mathbf{X}^C\right)\right), \tag{9}$$

$$\mathbf{X}^G = \text{FFN}\left(\text{Norm}\left(\mathbf{X}^{ST}\right)\right) + \mathbf{X}^{ST}. \tag{10}$$

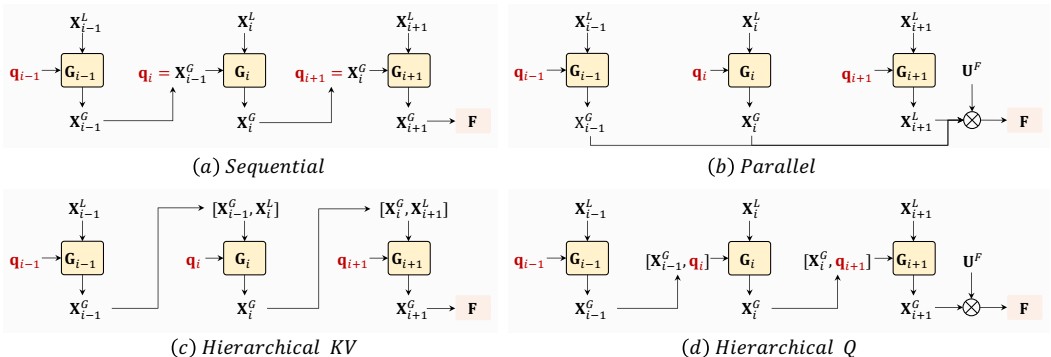

Figure 3: **Multi-Stage Fusion Block.**

The DPE is instantiated as depth-wise spatiotemporal convolution (Li et al., 2022a). We design the global C_MHRA in a cross-attention style to efficiently construct a video representation,

$$\mathrm{R}_n^{\mathrm{C}}(\mathbf{q}, \mathbf{X}) = \mathrm{A}_n^{\mathrm{C}}(\mathbf{q}, \mathbf{X})\mathrm{V}_n(\mathbf{X}), \tag{11}$$

$$\mathrm{C\_MHRA}(\mathbf{q}, \mathbf{X}) = \mathrm{Concat}(\mathrm{R}_1^{\mathrm{C}}(\mathbf{q}, \mathbf{X}); \mathrm{R}_2^{\mathrm{C}}(\mathbf{q}, \mathbf{X}); \cdots; \mathrm{R}_N^{\mathrm{C}}(\mathbf{q}, \mathbf{X}))\mathbf{U}. \tag{12}$$

$\mathrm{R}_n^{\mathrm{C}}(\mathbf{q}, \cdot)$ is the cross relation aggregator, which can convert a learnable query $\mathbf{q} \in \mathbb{R}^{1 \times C}$ into a video representation, via modeling dependency between this query $\mathbf{q}$ and all the spatiotemporal tokens $\mathbf{X}$. First, it computes the cross affinity matrix $\mathrm{A}_n^{\mathrm{C}}(\mathbf{q}, \mathbf{X})$ to learn relation between $\mathbf{q}$ and $\mathbf{X}$,

$$\mathrm{A}_n^{\mathrm{C}}(\mathbf{q}, \mathbf{X}_j) = \frac{\exp\{\mathrm{Q}_n(\mathbf{q})^T \mathrm{K}_n(\mathbf{X}_j)\}}{\sum_{j' \in \Omega_{T \times H \times W}} \exp\{\mathrm{Q}_n(\mathbf{q})^T \mathrm{K}_n(\mathbf{X}_{j'})\}}. \tag{13}$$

Then, it uses the linear projection to transform $\mathbf{X}$ as spatiotemporal context $\mathrm{V}_n(\mathbf{X})$. Subsequently, it aggregates such context $\mathrm{V}_n(\mathbf{X})$ into the learnable query, with guidance of their affinity $\mathrm{A}_n^{\mathrm{C}}(\mathbf{q}, \mathbf{X})$. Finally, the enhanced query tokens from all the heads are further fused as a final video representation, by linear projection $\mathbf{U} \in \mathbb{R}^{C \times C}$. Note the query token is zero-initialized for stable training.

**Discussion**. We further discuss the distinct design of our global UniBlock, compared to the one in the original UniFormer (Li et al., 2022a). **(I)** We add the global UniBlock on top of the local UniBlock, extracting multi-scale spatiotemporal representations in token form. Such design helps strengthen the discriminative video representation without compromising the pretrained architecture. **(II)** The typical global spatiotemporal attention is computationally heavy, due to its quadratic complexity. To pursue better accuracy-computation balance, we introduce a cross-attention style of global MHRA in UniFormerV2, thus largely reducing the computation complexity from $O(L^2)$ to $O(L)$, where $L$ is the number of tokens. More importantly, since the query $\mathbf{q}$ is learnable, it can adaptively integrate the spatiotemporal context from all $L$ tokens to boost video recognition. **(III)** The global UniBlock inherits DPE design from UniFormer, and we find it also helps in Table 9c.

### 3.3 MULTI-STAGE FUSION BLOCK

We propose a multi-stage fusion block to integrate all video tokens from each global UniBlock as in Fig. 3. For simplicity, we denote the $i$-th global block as $\mathbf{X}_i^G = \mathrm{G}_i(\mathbf{q}_i, \mathbf{X}_i^L)$. Given the tokens $\mathbf{X}_i^L$ from the local UniBlock, the global block transforms the learnable query $\mathbf{q}$ into a video token $\mathbf{X}_i^G$. In this paper, we explore four fusion strategies to integrate the video tokens from all the global blocks $\{\mathbf{X}_i^G\}_{i=1}^N$ into a final video representation $\mathbf{F}$, and employ the sequential way to conduct fusion regarding efficacy and efficiency.

The studied fusion methods are given below. **(a) Sequential:** We sequentially use the video token from the previous global block $\mathbf{X}_{i-1}^G$ as the query token in the current global block $\mathbf{q}_i$, where $\mathbf{X}_i^G = \mathrm{G}_i(\mathbf{X}_{i-1}^G, \mathbf{X}_i^L)$. **(b) Parallel:** We concatenate all the global tokens $\{\mathbf{X}_i^G\}_{i=1}^N$ in parallel, and use a linear projection $\mathbf{U}^F \in \mathbb{R}^{N \times C}$ to obtain the final token, where $\mathbf{F} = \mathrm{Concat}(\mathbf{X}_1^G, ..., \mathbf{X}_N^G)\mathbf{U}^F$. **(c) Hierarchical KV:** We use the video token from the previous global block $\mathbf{X}_{i-1}^G$ as a part of contextual tokens in the current global block, where $\mathbf{X}_i^G = \mathrm{G}_i(\mathbf{q}_i, [\mathbf{X}_{i-1}^G, \mathbf{X}_i^L])$. **(d) Hierarchical Q:** We use the video token from the previous global block $\mathbf{X}_{i-1}^G$ as a part of query tokens in the current global block, i.e., $\mathbf{X}_i^G = \mathrm{G}_i([\mathbf{X}_{i-1}^G, \mathbf{q}_i], \mathbf{X}_i^L)$.

| Method | Image Pretrain | | Video Pretrain | FT Epoch | Frame× Crop×Clip | Param. (M) | FLOPs (T) | K400 Top-1 | Top-5 |
|---|---|---|---|---|---|---|---|---|---|
| | Ready | Data | | | | | | | |
| SlowFast+NL (Feichtenhofer et al., 2019) | N/A | None | ✗ | 196 | 80×3×10 | 60 | 7.0 | 79.8 | 93.9 |
| X3D-XXL 312↑ (Feichtenhofer, 2020) | N/A | None | ✗ | 256 | 24×3×10 | 20 | 5.8 | 80.4 | 94.6 |
| UniFormerV1-B (Li et al., 2022a) | ✗ | IN-1K | ✗ | 110 | 32×3×4 | 50 | 3.1 | 83.0 | 95.4 |
| Swin-L 384↑ (Liu et al., 2022) | ✗ | IN-21K | ✗ | 30 | 32×5×10 | 200 | 105.4 | 84.9 | 96.7 |
| MViTv2-L 312↑ (Li et al., 2021) | ✗ | IN-21K | ✗ | 105 | 40×3×5 | 218 | 42.2 | 86.1 | 97.0 |
| TimeSformer-L (Bertasius et al., 2021) | ✔ | IN-21K | ✗ | 15 | 96×3×1 | 121 | 7.1 | 80.7 | 94.7 |
| Mformer-HR 336↑ (Patrick et al., 2021) | ✔ | IN-21K | ✗ | 35 | 16×3×10 | 109 | 28.8 | 81.1 | 95.2 |
| UniFormerV2-B/16 | ✔ | IN-21K | ✗ | 55 | 16×3×4 | 115 | 3.6 | 83.4 | 96.2 |
| UniFormerV2-L/16 | ✔ | IN-22K | ✗ | 55 | 32×3×2 | 355 | 12.2 | 85.4 | 97.0 |
| *Methods with web-scale data. WTS contains 60M unpublished video-text pairs. ALIGN contains 1.8B image-text pairs.* | | | | | | | | | |
| ViViT-H/14×2 (Arnab et al., 2021) | ✗ | JFT-300M | ✗ | 30 | 32×3×4 | 654 | 47.8 | 84.9 | 95.8 |
| TokenLearner-L/10 (Ryoo et al., 2021) | ✗ | JFT-300M | ✗ | 30 | 64×3×4 | 450 | 48.9 | 85.4 | 96.3 |
| MTV-H (Yan et al., 2022) | ✗ | JFT-300M | ✗ | 30 | 32×3×4 | 1000+ | 44.5 | 85.8 | 96.6 |
| Florence 384↑ (Yuan et al., 2021b) | ✗ | FLD-900M | ✗ | 30 | 32×3×4 | 647 | N/A | 86.5 | 97.3 |
| CoCa 576↑ (Yu et al., 2022) | ✗ | JFT-3B+ALIGN | ✗ | N/A | N/A | 1000+ | N/A | 88.9 | - |
| CoVeR 448↑ (Zhang et al., 2021a) | ✗ | JFT-300M | ✗ | 20† | 16×3×1 | 431 | 17.6 | 86.3 | - |
| CoVeR 448↑ (Zhang et al., 2021a) | ✗ | JFT-3B | ✗ | 20† | 16×3×1 | 431 | 17.6 | 87.1 | - |
| MTV-H (Yan et al., 2022) | ✗ | IN-21K | WTS-60M | 30 | 32×3×4 | 1000+ | 44.5 | 89.1 | 98.2 |
| MTV-H 280↑ (Yan et al., 2022) | ✗ | IN-21K | WTS-60M | 30 | 32×3×4 | 1000+ | 73.6 | 89.9 | 98.3 |
| EVL-L/14 (frozen) 336↑ (Lin et al., 2022) | ✔ | CLIP-400M | ✗ | 53 | 32×3×1 | 67 | 19.1 | 87.7 | - |
| X-CLIP-L/14 336↑ (Ni et al., 2022) | ✔ | CLIP-400M | ✗ | 30 | 16×3×4 | 453 | 37.0 | 87.7 | - |
| UniFormerV2-L/14 (frozen) 336↑ | ✔ | CLIP-400M | K710-0.66M | 5 | 8×1×3 | 51 | 4.7 | 87.8 | 98.0 |
| UniFormerV2-L/14 (frozen) 336↑ | ✔ | CLIP-400M | K710-0.66M | 5 | 32×3×1 | 51 | 18.8 | 88.8 | 98.1 |
| UniFormerV2-B/16 | ✔ | CLIP-400M | K710-0.66M | 5 | 8×1×3 | 115 | 0.4 | 85.2 | 96.7 |
| UniFormerV2-B/16 | ✔ | CLIP-400M | K710-0.66M | 5 | 8×3×4 | 115 | 1.8 | 85.6 | 97.0 |
| UniFormerV2-L/14 | ✔ | CLIP-400M | K710-0.66M | 5 | 8×3×4 | 354 | 8.0 | 88.8 | 98.1 |
| UniFormerV2-L/14 | ✔ | CLIP-400M | K710-0.66M | 5 | 32×3×2 | 354 | 16.0 | 89.3 | 98.2 |
| UniFormerV2-L/14 336↑ | ✔ | CLIP-400M | K710-0.66M | 5 | 32×3×2 | 354 | 37.6 | 89.7 | 98.3 |
| UniFormerV2-L/14 336↑ | ✔ | CLIP-400M | K710-0.66M | 5 | 64×3×2 | 354 | 75.3 | **90.0** | **98.4** |

Table 1: **Comparison with the state-of-the-art on Kinetics-400.** FT indicates the video finetuning. † marks co-finetuning with K400+SSV2+MiT+IN-1K. Our UniFormerV2 outperforms most of the current methods in terms of accuracy and/or efficiency. It firstly achieves **90.0% top-1 accuracy** on K400. More explanations of model comparison can be found in the text.

| Method | Frame×Crop×Clip | Param. (M) | FLOPs (T) | K600 Top-1 | Top-5 | K700 Top-1 | Top-5 |
|---|---|---|---|---|---|---|---|
| SlowFast+NL (Feichtenhofer et al., 2019) | 80×3×10 | 60 | 7.0 | 81.8 | 95.1 | 71.0 | 89.6 |
| MoViNet-A6 (Kondratyuk et al., 2021) | 120×1×1 | 31 | 0.4 | 83.5 | 96.2 | 72.3 | - |
| MViTv2-L 312↑ (Li et al., 2021) | 40×3×3 | 218 | 42.2 | 87.5 | 97.8 | 79.4 | 94.9 |
| CoVeR 448↑ (Zhang et al., 2021a) | 16×3×1 | 431 | 431 | 87.9 | - | 79.8 | - |
| MTV-H (Yan et al., 2022) | 32×3×4 | 1000+ | 44.5 | 89.6 | 98.3 | 82.2 | 95.7 |
| CoCa 576↑ (Yu et al., 2022) | N/A | 1000+ | N/A | 89.4 | - | **82.7** | - |
| UniFormerV2-L/14 (frozen) 336↑ | 32×3×1 | 51 | 18.8 | 89.1 | 98.2 | 80.6 | 95.2 |
| UniFormerV2-L/14 | 32×3×2 | 354 | 16.0 | 89.5 | 98.3 | 81.5 | 95.7 |
| UniFormerV2-L/14 336↑ | 32×3×2 | 354 | 37.6 | 89.9 | 98.5 | 82.1 | 96.1 |
| UniFormerV2-L/14 336↑ | 64×3×2 | 354 | 75.3 | **90.1** | **98.5** | **82.7** | **96.2** |

Table 2: **Comparison with the state-of-the-art on Kinetics-600/700.**

Finally, we dynamically integrate the *final* tokens from both local and global blocks, effectively promoting recognition performance in empirical studies (Table 12). Specifically, we extract the class token $\mathbf{F}^C$ from the final local UniBlock, and add it with the video token $\mathbf{F}$ by weighted sum, i.e., $\mathbf{Z} = \alpha \mathbf{F} + (1 - \alpha)\mathbf{F}^C$, where $\alpha$ is a learnable parameter processed by the Sigmoid function.

# 4 EXPERIMENTS

**Datasets**. To verify the learning capacity of our UniFormerV2, we conduct experiments on 8 popular video benchmarks, including the *trimmed* videos less than 10 seconds, and the *untrimmed* videos more than 1 min. For the trimmed video benchmarks, we divide them into two categories. (a) Scene-related datasets: *Kinetics* family (Kay et al., 2017) (i.e., Kinetics-400, 600 and 700) and *Moments in Time* V1 (Monfort et al., 2020). (b) Temporal-related datasets: *Something-Something* V1/V2 (Goyal et al., 2017b). For the untrimmed video recognition, we choose *ActivityNet* (Heilbron et al., 2015) and *HACS* (Zhao et al., 2019). More dataset details can be found in Appendix A.

**Kinetics-710 for Post-Pretraining** We propose a unified video benchmark for post-pretraining UniFormerV2. Different from (Yan et al., 2022) that exploits a web-scale video dataset (i.e., 60M video-text pairs), we build a much smaller video benchmark based on the Kinetics-400/600/700.

| Method | Modality | ViT | Image Pretrain | | Frame× | Param. | FLOPs | MiT V1 | |
|---|---|---|---|---|---|---|---|---|---|
| | | | Ready | Data | Crop×Clip | (M) | (T) | Top-1 | Top-5 |
| AssembleNet-101 (Ryoo et al., 2020) | RGB+Flow | ✗ | N/A | None | N/A | 53 | 0.8 | 34.3 | 62.7 |
| MoViNet-A6 (Kondratyuk et al., 2021) | RGB | ✗ | N/A | None | 120×1×1 | 31 | 0.3 | 40.2 | - |
| ViViT-L/16×2 FE (Arnab et al., 2021) | RGB | ✓ | ✓ | IN-21K | 32×3×1 | 612 | 11.9 | 38.5 | 64.2 |
| MTV-H (Yan et al., 2022) | RGB | ✓ | ✗ | IN-21K | 32×3×4 | 1000+ | 44.5 | 45.6 | 74.7 |
| MTV-H 280↑ (Yan et al., 2022) | RGB | ✓ | ✗ | IN-21K | 32×3×4 | 1000+ | 73.6 | 47.2 | 75.7 |
| CoVeR 448↑ (Zhang et al., 2021a) | RGB | ✓ | ✗ | JFT-3B | 16×3×1 | 431 | 17.6 | 46.1 | - |
| UniFormerV2-B/16 | RGB | ✓ | ✓ | CLIP-400M | 8×3×4 | 115 | 1.8 | 42.7 | 71.5 |
| UniFormerV2-L/14 | RGB | ✓ | ✓ | CLIP-400M | 8×3×4 | 354 | 8.0 | 47.0 | 76.1 |
| UniFormerV2-L/14 336↑ | RGB | ✓ | ✓ | CLIP-400M | 8×3×4 | 354 | 18.8 | **47.8** | **76.9** |

Table 3: **Comparison with the state-of-the-art on Moments in Time V1.**

| Method | Image Pretrain | | Video Pretrain | | FT | Frame× | Param. | FLOPs | SSV2 | |
|---|---|---|---|---|---|---|---|---|---|---|
| | Ready | Data | Data | Epoch | Epoch | Crop×Clip | (M) | (T) | Top-1 | Top-5 |
| VideoMAE-B (Tong et al., 2022) | N/A | None | SSV2 | 2400 | 40 | 16×3×2 | 87 | 1.1 | 70.3 | 92.7 |
| VideoMAE-L (Tong et al., 2022) | N/A | None | SSV2 | 2400 | 40 | 32×3×1 | 305 | 4.3 | 75.3 | 95.2 |
| MViTv1-B (Fan et al., 2021) | N/A | None | K400 | 200 | 100 | 64×3×1 | 36.6 | 1.4 | 67.7 | 90.9 |
| MaskFeat-L 312↑ (Wei et al., 2022) | N/A | None | K400 | 905 | 40 | 40×3×4 | 218 | 28.3 | 74.4 | 94.6 |
| MViTv2-B (Li et al., 2021) | ✗ | IN-21K | K400 | 100 | 100 | 32×3×1 | 51.1 | 0.7 | 72.1 | 93.4 |
| UniFormerV1-B (Li et al., 2022a) | ✗ | IN-1K | K400 | 110 | 50 | 32×3×1 | 50 | 0.8 | 71.2 | 92.8 |
| Swin-B (Liu et al., 2022) | ✗ | IN-21K | K400 | 30 | 60 | 32×3×1 | 89 | 1.0 | 69.6 | 92.7 |
| CoVeR 448↑ (Zhang et al., 2021a) | ✗ | JFT-3B | None | 0 | 20† | 16×3×1 | 431 | 17.6 | 70.8 | - |
| ViViT-L/16×2 FE (Arnab et al., 2021) | ✓ | IN-21K | K400 | 30 | 35 | 32×3×14 | 612 | 47.6 | 65.4 | 89.8 |
| MTV-B 320↑ (Yan et al., 2022) | ✓ | IN-21K | K400 | 30 | 100 | 32×3×4 | 310 | 11.2 | 68.5 | 90.4 |
| TimeSformer-HR (Bertasius et al., 2021) | ✓ | IN-21K | None | 0 | 15 | 16×3×1 | 121 | 5.1 | 62.5 | - |
| EVL-L/14 (Lin et al., 2022) | ✓ | CLIP-400M | None | 0 | 46 | 32×3×1 | 67 | 9.6 | 66.7 | - |
| UniFormerV2-B/16 | ✓ | CLIP-400M | None | 0 | 22 | 16×3×1 | 163 | 0.6 | 69.5 | 92.3 |
| UniFormerV2-B/16 | ✓ | CLIP-400M | None | 0 | 22 | 32×3×1 | 163 | 1.1 | 70.7 | 93.2 |
| UniFormerV2-L/14 | ✓ | CLIP-400M | None | 0 | 15 | 16×3×1 | 574 | 2.6 | 72.1 | 93.6 |
| UniFormerV2-L/14 | ✓ | CLIP-400M | None | 0 | 15 | 32×3×1 | 574 | 5.2 | **73.0** | **94.5** |

Table 4: **Comparison with the state-of-the-art on Something-Something V2.** The methods without image pretraining are marked in gray. † marks co-finetuning with K400+SSV2+MiT+IN-1K.

Concretely, we merge the training set of these Kinetics datasets, and then delete the repeated videos according to Youtube IDs. Note we also remove testing videos from different Kinetics datasets leaked in our combined training set for correctness. As a result, the total number of training videos is reduced from 1.14M to 0.66M. Additionally, we merge the action categories in these three Kinetics datasets, which leads to 710 classes in total. Hence, we call this video benchmark Kinetics-710. More detailed descriptions can be found in Appendix F. In our experiments, we empirically show the effectiveness of our Kinetics-710. For post-pretraining, we simply use 8 input frames and adopt the same hyperparameters as training on the individual Kinetics dataset. After that, no matter how many frames are input (16, 32, or even 64), we only need 5-epoch finetuning for more than 1% top-1 accuracy improvement on Kinetics-400/600/700, as shown in Table 9e.

**Implement Details**. Unless stated otherwise, we follow most of the training recipes in UniFormer (Li et al., 2022a), and the detailed training hyperparameters can be found in Appendix A. We build UniFormerV2 based on ViTs pretrained with various supervisions (see Table 8), showing the generality of our design. For the best result, we adopt CLIP-ViT (Radford et al., 2021) as the backbone by default, due to its robust representation pretrained by vision-language contrastive learning. For most datasets, we insert the global UniBlocks in the last 4 layers of ViT-B/L to perform the multi-stage fusion. But for Sth-Sth V1/V2, we insert the global UniBlocks in the last 8/16 layers of ViT-B/L for better temporal modeling. The corresponding ablation studies are shown in Table 9. Finally, we adopt sparse sampling (Wang et al., 2016) with the resolution of 224 for all the datasets.

## 4.1 COMPARISON TO STATE-OF-THE-ART

**Kinetics.** Table 1 presents the state-of-the-art comparison on Kinetics-400. (1) The first part lists the models pretrained on open-source datasets like ImageNet (Deng et al., 2022). On one hand, compared with UniFormerV1-B (Li et al., 2022a), our UniFormerV2-B only uses 50% fine-tuning epochs but achieves a better accuracy, showing the importance of inheriting the pretrained weights. On the other hand, compared with TimeSformer-L (Bertasius et al., 2021), our model achieves 2.7% performance gain with 50% FLOPs, showing the importance of adopting the UniFormer designs. Besides, compared with Swin-L (Liu et al., 2022), our UniFormerV2-L based on BeiT (Bao et al., 2021) that pretrained on ImageNet-22K, achieves comparable results but with 12% FLOPs. (2) The second part shows the methods using web-scale data. On one hand, compared with MTV-H

| Method | Frame | Top-1 | Top-5 |
|---|---|---|---|
| TSN-R50(Wang et al., 2016) | 16 | 19.9 | 47.3 |
| TSM-R50(Lin et al., 2019) | 16 | 47.2 | 77.1 |
| TEA-R50 (Li et al., 2020b) | 16 | 51.9 | 80.3 |
| CT-Net-R50 (Li et al., 2020a) | 16 | 52.5 | 80.9 |
| TDN-R101 (Wang et al., 2021a) | 16 | 55.3 | 88.3 |
| UniFormerV1-S (Li et al., 2022a) | 16 | 57.1 | 84.9 |
| UniFormerV1-B (Li et al., 2022a) | 32 | 61.0 | 87.6 |
| UniFormerV2-B/16 | 16 | 56.8 | 84.2 |
| UniFormerV2-B/16 | 32 | 59.4 | 86.2 |
| UniFormerV2-L/14 | 16 | 60.5 | 86.5 |
| UniFormerV2-L/14 | 32 | **62.7** | **88.0** |

Table 5: **Results on Something-Something V1.**

| Method | Frame | Top-1 |
|---|---|---|
| DSN-R34 (Zheng et al., 2020) | 32 | 82.6 |
| MARL-R152 (Wu et al., 2019) | 32 | 85.7 |
| NSNet-Swin-L (Xia et al., 2022) | 32 | 90.2 |
| UniFormerV2-L/14 | 16 | 94.3 |
| UniFormerV2-L/14 | 32 | **94.7** |

Table 6: **Results on ActivityNet.**

| Method | Frame | Top-1 |
|---|---|---|
| CSN-R152 (Tran et al., 2019) | 32 | 91.5 |
| TimeSformer (Bertasius et al., 2021) | 8 | 91.8 |
| ViViT-B (Arnab et al., 2021) | 32 | 91.9 |
| UniFormerV2-L/14 | 16 | **95.5** |
| UniFormerV2-L/14 | 32 | 95.4 |

Table 7: **Results on HACS.**

| Type | Method | Data | K400 | SSV2 |
|---|---|---|---|---|
| | TimeSformer | IN-21K | 78.7 | 59.5 |
| SL | ViT | IN-21K | 81.6 | 67.5 |
| | DeiT III | IN-21K | 82.7 | 66.5 |
| CL | DINO | IN-1K | 78.7 | 65.8 |
| | CLIP | CLIP-400M | **84.4** | **69.5** |
| MIM | MAE | IN-1K | 78.8 | 65.1 |
| | BeiT | IN-22K | 82.2 | 67.7 |

Table 8: **Different pretrained ViTs**. Our UniFormerV2 based on different open-sourced ViTs beat TimeSformer, especially for Something-Something.

(ensembling 4 models) (Yan et al., 2022), our single model only requires 1% video post-pretraining, 16% finetuning epochs and 35% model parameters to achieve competitive accuracy. On the other hand, under the same CLIP-400M pretraining, our UniFormerV2-L (frozen) only uses 25% FLOPs to achieve the competitive accuracy compared with EVL-L (frozen) (Lin et al., 2022), and obtains 1.1% accuracy improvement with similar FLOPs. Finally, our UniFormerV2 is the first model to achieve **90.0%** top-1 accuracy on K400, to our best knowledge. For Kinetics-600 and 700, our UniFormerV2 also obtains the state-of-the-art performance (90.1% and 82.7%, see Table 2).

**Moments in Time.** Due to complex inter-class and intra-class variation, MiT is more challenging than Kinetics. As shown in Table 3, our model beats most of the recent methods, i.e., compared with ViViT-L (Arnab et al., 2021), UniFormerV2-B obtains 4.2% performance gain but only with 19% model parameters and 15% FLOPs. Compared with MTV-H (Yan et al., 2022), UniFormerV2-L only uses 35% model parameters and 25% FLOPs to achieve 1.2% top-5 accuracy improvement.

**Something-Something.** In Table 4, we show the results on Sth-SthV2. First, our model outperforms those standard models based on the well-pretrained image ViT on hand. For example, under the same CLIP-400M pretraining and the same number of sampled frames, our UniFormerV2-B obtains 4% higher accuracy with only 11% FLOPs, compared with EVL-L (Lin et al., 2022). Second, we compare our model with those models whose backbone is specially designed. Since the pretraining is unavailable for these models, they have to perform a tedious training phrase, consisting of image-pretraining, video pretraining and video finetuning. Alternatively, our UniFormerV2 can work well with only video finetuning, e.g., our model only uses 22 epochs to achieve the performance of UniFormerV1 (Li et al., 2022a), which requires 110+50=160 video epochs to obtain results. Finally, we compare UniFormerV2 with those models which do not apply image pretraining. Such models require a huge number of training epochs, e.g., VideoMAE-B (Tong et al., 2022) contains 2400 video pretraining epochs and 40 video finetuning epochs, much longer than our UniFormerV2-B with a similar accuracy (only 22 video finetuning epochs, i.e., 0.9 % training epochs of VideoMAE-B). For Sth-Sth V1 in Table 5, we reach the new state-of-the-art performance (**62.7%**). The above results reveal the effectiveness and efficiency of our UniFormerV2 for temporal modeling.

**ActivityNet and HACS.** For the untrimmed videos, it is essential to capture long-range temporal information, since the action may occur multiple times at arbitrary moments. As shown in Table 6 and 7, our UniFormerV2 significantly outperforms the previous best results on the large-scale untrimmed benchmark ActivityNet and HACS by **4.5% and 3.6%**, respectively. These results demonstrate the strong long-term modeling capacity of our UniFomrerV2.

## 4.2 ABLATION STUDIES

To evaluate the effectiveness of UniFormerV2, we investigate each key structure design, as shown in Table 8 and Table 9. All the models are directly finetuned from CLIP-ViT-B/16 by default. We utilize '8×4×3' and '16×1×3' testing strategies for Kinetics and Something-Something respectively.

| Global | Local | T-Down | K400 | SSV2 |
|--------|-------|--------|------|------|
| ✗ | ✗ | ✗ | 83.1 | 45.1 |
| ✔ | ✗ | ✗ | **84.4** | 63.3 |
| ✗ | ✔ | ✗ | 83.6 | 67.7 |
| ✔ | ✔ | ✗ | **84.4** | 68.7 |
| ✔ | ✔ | ✔ | **84.4** | **69.5** |

(a) **Components of UniFormerV2.**

| Design | SSV2 |
|--------|------|
| Temporal MHSA | 65.2 |
| Temporal Convolution | 67.5 |
| ST-Adapter | 68.0 |
| Local MHRA | 69.1 |
| Local MHRA + DPE | 69.1 |
| Local MHRA × 2 | **69.5** |

| Layers | Reduction | SSV2 |
|--------|-----------|------|
| 1-12 | 4.0 | 68.9 |
| 1-12 | 2.0 | 69.1 |
| 1-12 | 1.5 | **69.5** |
| 1-12 | 1.0 | **69.5** |
| 1-8 | 1.5 | 67.9 |
| 1-4 | 1.5 | 67.6 |

(b) **Local UniBlock.**

| Layers | DPE | K400 | SSV2 |
|--------|-----|------|------|
| 9-12 | ✗ | 84.2 | 68.1 |
| 9-12 | ✔ | 84.4 | 68.5 |
| 5-12 | ✔ | 84.4 | **69.5** |
| 1-12 | ✔ | 84.4 | 69.4 |

(c) **Global UniBlock.**

| Design | SSV2 |
|--------|------|
| Sequential | **69.5** |
| Parallel | 69.1 |
| Hierarchical KV | 68.9 |
| Hierarchical Q | **69.5** |

(d) **Multi-Stage Fusion.**

| Pretraining | Finetuning | K400 | K600 | K700 |
|-------------|------------|------|------|------|
| None | Individual | 84.4 | 85.0 | 75.8 |
| K400/600/700 | K400/600/700 | **85.6** | 86.0 | 75.6 |
| K710 | K400/600/700 | **85.6** | **86.3** | 76.1 |
| K710 | Individual | **85.6** | **86.3** | **76.3** |

(e) **Different Training Scripts.**

Table 9: **Ablation studies.** T-Down means temporal downsampling, and we double the frames to maintain similar GFLOPs. ST-Adapter is proposed in Pan et al. (2022). Compared with simple co-training, our K710 pretraining saves 33% cost with consistent improvement (see Appendix A).

**Pretraining Sources.** To demonstrate the generality of our UniFormerV2 design, we apply it on the ViTs with different pertaining methods, including supervised learning (Dosovitskiy et al., 2021; Touvron et al., 2022), contrastive learning(Caron et al., 2021; Radford et al., 2021) and mask image modeling (He et al., 2022; Bao et al., 2021). Table 8 shows that all the models beat TimeSformer (Bertasius et al., 2021), especially for Something-Something that relies on strong temporal modeling. It also reflects that a better-pretrained ViT is helpful for stronger video performance.

**Different Components.** In Table 9a, note the global UniBlock is crucial for the scene-related benchmark (e.g., K400), since this block can effectively provide holistic video representation for classification. Alternatively, the local UniBlock is critical for the temporal-related benchmark (e.g., SSV2), since this block can effectively describe detailed video representation for classification. Besides, using temporal downsampling with double input frames (similar FLOPs) is also helpful for distinguishing fine-grained videos like SSV2, due to the larger temporal receptive field.

**Local UniBlock.** To explore the structure of local UniBlock, we conduct experiments in Table 9b. It reveals that convolution is better than self-attention for temporal modeling, and our local MHRA is more powerful than both of them in SSV2. Following ST-Adapter (Pan et al., 2022), we add another local MHRA after the spatial MHRA for better performance. Besides, we add local MHRA in all the layers and reduce the channel by 1.5 times for the best accuracy-flops trade-off.

**Global UniBlock and Multi-stage Fusion.** In Table 9c, we find that the features in the deep layers are critical for capturing long-term dependency, while the DPE and the middle information are necessary for identifying the motion difference. For the fusion strategy, Table 9d shows that the simplest sequential fusion is adequate for integrating multi-stage features.

**Training Recipes.** We compare different training and finetuning methods in Table 9e. Note that when co-training with K400, K600 and K700, we remove the leaked videos in the validation set and introduce three classification heads. K710 maintains only about 60% of the total training videos (0.66M vs. 1.14M for K400+K600+K700), but it improves classification performance significantly for Kinetics. Meanwhile it saves about 33% training cost (see Appendix A). Besides, direct training on it works better than a Kinetics co-training, especially for K600 (+1.3% vs. +1.0%) and K700 (+0.5 vs. -0.2%). Though co-finetuning shared the backbone and saved parameters, we adopt individual finetuning for each dataset considering the best performance.

# 5 CONCLUSION

In this paper, we propose a powerful video model, namely UniFormerV2. It arms image-pretrained ViTs with efficient UniFormer designs for video learning. By novel local and global video relation aggregators, it is capable of conducting effective spatiotemporal modeling with a tractable complexity. Besides of seamlessly integrating advantages from both ViTs and UniFormer, we also introduce multi-scale token fusion for further enhancing video representation. Our UniFormerV2 achieves state-of-the-art performance on 8 popular video benchmarks, and firstly reaches 90% top-1 accuracy on Kinetics-400, to our best knowledge.

**Reproducibility.** To ensure all the results can be reproduced, we give the details of the datasets, model and training hyperparameters in our experiments (see Table 10 and Table 11). For Kinetics-710, we provide its label list in Table 20 for reproduction. All the codes are based on the UniFormer (Li et al., 2022b) repository.

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

| Dataset | Training #Samples | Validation #Samples | Average Length | #Actions |
|---|---|---|---|---|
| Kinetics-710 (ours) | 658,340 | 66,803 | 10s | 710 |
| Kinetics-400 (Carreira & Zisserman, 2017) | 240,436 
 246,245* | 19,787 
 20,000* | 10s | 400 |
| Kinetics-600 (Carreira et al., 2018) | 366,006 
 392,622* | 27,935 
 30,000* | 10s | 600 |
| Kinetics-700 (Carreira et al., 2019) | 529,573 
 545,317* | 33,861 
 35,000* | 10s | 700 |
| Moments in Time (Monfort et al., 2020) | 802,244 
 802,264* | 33,899 
 33,900* | 3s | 339 |
| Something-Something V1 (Goyal et al., 2017b) | 86,017 | 11,522 | 4.0s | 174 |
| Something-Something V2 (Goyal et al., 2017b) | 168,913 | 24,777 | 4.0s | 174 |
| ActivityNet (Heilbron et al., 2015) | 10,024 | 4,926 | 117s | 200 |
| HACS (Zhao et al., 2019) | 37,452 | 5,953 | 149s | 200 |

Table 10: **Dataset descriptions.** * indicates the original video number.

| | K710 | K400/600/700 | MiT | ANet&HACS | SSV1/V2 |
|---|---|---|---|---|---|
| *Optimization* | | | | | |
| Optimizer | | AdamW (Loshchilov & Hutter, 2017a) | | | |
| Momentum | | $\beta_1, \beta_2 = 0.9, 0.999$ | | | |
| Weight decay | | 0.05 | | | |
| Learning rate schedule | | cosine decay (Loshchilov & Hutter, 2017b) | | | |
| Start learning rate | | 1e-6 | | | |
| End learning rate | | 1e-6 | | | |
| Batch size | 512 | 256 | 512 | 64 | 128 |
| Learning rate (Base) | 2e-5 | 2e-6 | 2e-5 | - | 4e-5 |
| Learning rate (Large) | 1e-5 | 1.5e-6 | 1e-5 | 5e-6 | 2e-5 |
| Warmup epochs (Goyal et al., 2017a) | 5 | 1 | 5 | 5 | 5 |
| Total epochs (Base) | 55 | 5 | 24 | - | 22 |
| Total epochs (Large) | 40 | 5 | 18 | 20 | 15 |
| *Data augmentation* | | | | | |
| Inception-style cropping Szegedy et al. (2015) | | | | | |
|    Scale | | [0.08, 1.00] | | | |
|    Jitter aspect ratio | | [0.75, 1.33] | | | |
| Color jitter | | 0.4 | | | |
| Rand augment (Cubuk et al., 2020) | | rand-m7-n4-mstd0.5-inc1 | | | |
| Repeated sampling (Hoffer et al., 2020) | 1 | 1 | 1 | 2 | 2 |
| *Regularisation* | | | | | |
| Dropout (Srivastava et al., 2014) | | | | | |
|    Backbone | | | 0.5 | | |
|    Global branch | | | 0.5 | | |
| Drop path (Huang et al., 2016) | | | | | |
|    Backbone | - | - | - | 0.2 | 0.2 |
|    Global branch | - | - | - | 0.4 | 0.4 |

Table 11: **Training hyperparameters for our experiments.** "-" indicates that the related method is not used. Values constant in all the datasets are listed once. Datasets are denoted as follows: K (Kinetics), MiT (Moments in Time), ANet (ActivityNet), SS (Something-Something).

## A ADDITIONAL IMPLEMENTATION DETAILS

**Datasets.** In Table 10, we give more details of our datasets. *Kinetics* family (Kay et al., 2017) is the most widely-used benchmark, includes Kinetics-400, 600 and 700. Since some videos are unavailable on YouTube, the Kinetics datasets are gradually shrinking over time. We report the video number of our version for a more fair comparison. *Moments in Time* V1 (Monfort et al., 2020) contains 0.8M 3-second video clips annotated with 339 classes, which suggests capturing the gist of a dynamic scene. *Something-Something* V1/V2 Goyal et al. (2017b) consist of 174 actions interacted with everyday objects. They require strong temporal modeling to distinguish confusing actions such as opening/closing something. *ActivityNet* (Heilbron et al., 2015) and *HACS* (Zhao et al., 2019) are two large-scale untrimmed video benchmark. They respectively contain about 20K and 50K videos in 200 human daily living actions. For these two datasets, we sample those video clips containing

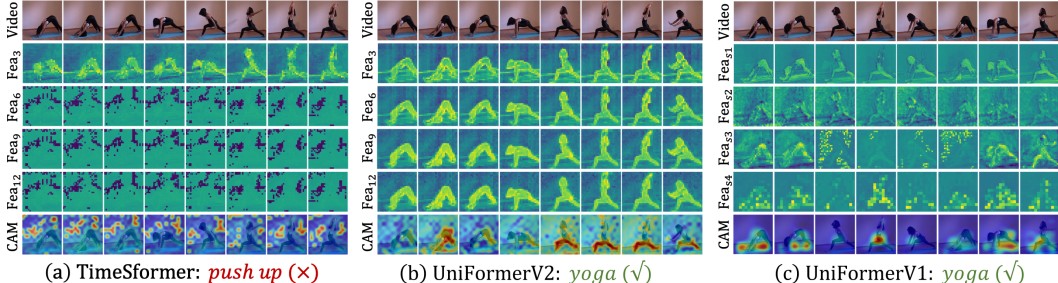

Figure 4: **More visualizations.** Frames are sampled from Kinetics according to different sampling strategies in different methods. For UniFormerV1, it samples double frames and downsamples the temporal resolution in the patch embedding.

action for training, thus we do not add another background class. While for testing, we sample the frames sparsely from the whole untrimmed videos.

**Implementation Details.** For the scene-related datasets, we only insert the global UniBlocks in the last 4 layers of ViT-B/L to perform multi-stage fusion, since the local UniBlocks and temporal downsampling do not further improve the results in Table 9a. But for Something-Something V1/V2, we adopt all the designs and insert the global UniBlocks in the last 8/16 layers of ViT-B/L for better temporal modeling. Besides, when finetuning those models with large-scale dataset pretraining, it is necessary to initialize the new parameters properly. For stable training, we zero initialize some of the layers, including the last point-wise convolutions in the local temporal MHRA, the query tokens and output projection layers in the query-based cross MHRA, the last linear layers in the FFN of the global UniBlock, and the learnable fusion weights. What's more, we provide the detailed hyperparameters in Table 11. Most of the training scripts follow UniFormer (Li et al., 2022a), but differently, we do not apply Mixup (Zhang et al., 2018), CutMix (Yun et al., 2019), Label Smoothing (Szegedy et al., 2016) and Random Erasing (Zhong et al., 2020). When finetuning the full models on Kinetics directly from image pretraining, we adopt the same hyperparameters as in K710 pretraining. If the backbone is frozen, we use a larger learning rate (4e-4) without warmup.

**Training Cost.** In table 9e, we compare different training scripts. When finetuning Kinetics-400, 600 and 700 individually, we train the models for 55 epochs, and the total training data is about $0.24 + 0.366 + 0.529 \approx 1.14\text{M}$. When pretraining with Kinetics-710 (0.66M), we only finetune the models for 5 epochs. Thus the percentage of saving cost is as follows,

$$1 - \frac{0.66 \times 55 + 1.14 \times 5}{1.14 \times 55} \approx 0.33 \tag{14}$$

Thus we save almost 33% of the training cost. More importantly, for the models with more frames (16, 32, or even 64), we only need to finetune the K710 pretrained models with 8 frames. Our training scripts are very efficient while effective for the Kinetics family.

# B  VISUALIZATIONS

In Figure 4, we compared UniFormerV2 with the typical ViT-based model, i.e., TimeSformer (Bertasius et al., 2021), and UniFormerV1 (Li et al., 2022a) through visualization. Since UniFormerV1 is a multi-scale architecture, we show its features at the bottom of 4 stages. For TimeSformer and UniFormerV2, they are based on ViTs with a fixed resolution, thus we show their features every 3 layers. We use CAM (Zhou et al., 2016) to show the most discriminative features that the network locates. The red parts indicate where the models focus more on, while the blue parts are ignored.

It reveals that both UniFormerV1 and UniFormerV2 are good at capturing local details, but UniFormerV1 may lose information in deeper layers due to the shrinking resolution, thus it fails to activate the discriminative parts. In contrast, TimeSformer only learns local features in the shallow layers, thus struggling to focus on meaningful areas. As for UniFormerV2, it surprisingly maintains local details even in the deep layers. More importantly, it can observe the whole video and learn to concentrate more on the woman's leg, which helps recognize the action. These results demonstrate that our UniFormerV2 is effective to capture local details and long-term dependency.

| Method | #Frame | K400 |
|---|---|---|
| Only Global | 8×3×4 | 81.8 |
| Local+Global | 8×3×4 | **84.4** |

Table 12: **Output token combination.**

| Pretrain | #Frame | SSV1 | SSV2 |
|---|---|---|---|
| CLIP-400M | 16×3×1 | **56.8** | **69.5** |
| CLIP-400M+K400 | 16×3×1 | 55.8 | 68.4 |

Table 13: **K400 pretraining.**

| #Query | #Frame | SSV2 |
|---|---|---|
| 1 | 16×3×1 | **69.5** |
| 4 | 16×3×1 | 69.1 |
| 16 | 16×3×1 | 68.6 |

Table 14: **Query Number.**

| Method | Param. (M) | FLOPs (G) | SSV2 |
|---|---|---|---|
| Mean Pooling | 86 | 422 | 45.1 |
| Divided Space-Time MHSA | 114 | 555 | 63.4 |
| Joint Space-Time MHSA | 86 | 539 | 65.8 |
| Temporal Convolution | 86 | 422 | 65.6 |
| Temporal Shift | 86 | 422 | 65.7 |
| Temporal Transformer | 128 | 423 | 61.5 |
| Local MHRA (Ours) | 105 | 511 | **67.7** |

Table 15: **Different modules.**

| Method | Pretrain | Frame× Crop×Clip | Param. (M) | FLOPs (T) | K400 Top-1 | K400 Top-5 | K600 Top-1 | K600 Top-5 | K700 Top-1 | K700 Top-5 |
|---|---|---|---|---|---|---|---|---|---|---|
| UniFormerV2-B/16 | | 8×1×3 | 115 | 0.4 | 84.0 | 96.3 | 84.8 | 96.8 | 75.4 | 92.6 |
| UniFormerV2-B/16 | CLIP-400M | 8×3×4 | 115 | 1.6 | 84.4 | 96.3 | 85.0 | 97.0 | 75.8 | 92.8 |
| UniFormerV2-L/14 | | 8×1×3 | 354 | 2.0 | 87.3 | 97.7 | 87.8 | 97.6 | 80.0 | 95.0 |
| UniFormerV2-L/14 | | 8×3×4 | 354 | 8.0 | 87.7 | 98.3 | 88.0 | 97.7 | 80.3 | 95.2 |
| UniFormerV2-B/16 | | 8×1×3 | 115 | 0.4 | 85.2 | 96.7 | 85.6 | 97.0 | 75.8 | 92.4 |
| UniFormerV2-B/16 | | 8×3×4 | 115 | 1.8 | 85.6 | 97.0 | 86.1 | 97.2 | 76.3 | 92.7 |
| UniFormerV2-L/14 | | 8×1×3 | 354 | 2.0 | 88.4 | 97.9 | 88.6 | 98.1 | 80.4 | 95.2 |
| UniFormerV2-L/14 | | 8×3×4 | 354 | 8.0 | 88.8 | 98.1 | 89.0 | 98.2 | 80.8 | 95.4 |
| UniFormerV2-L/14 | | 16×3×1 | 354 | 4.0 | 88.9 | 98.0 | 89.2 | 98.2 | 80.9 | 95.4 |
| UniFormerV2-L/14 | CLIP-400M +K710 | 16×3×4 | 354 | 16.0 | 89.1 | 98.2 | 89.4 | 98.3 | 81.2 | 95.6 |
| UniFormerV2-L/14 | | 32×3×1 | 354 | 16.0 | 89.2 | 98.2 | 89.3 | 98.2 | 81.3 | 95.6 |
| UniFormerV2-L/14 | | 32×3×2 | 354 | 16.0 | 89.3 | 98.2 | 89.5 | 98.3 | 81.5 | 95.7 |
| UniFormerV2-L/14 | | 32×3×4 | 354 | 32.0 | 89.5 | 98.2 | 89.5 | 98.3 | 81.4 | 95.8 |
| UniFormerV2-L/14 336↑ | | 32×3×2 | 354 | 37.6 | 89.7 | 98.3 | 89.9 | 98.5 | 82.1 | 96.1 |
| UniFormerV2-L/14 336↑ | | 32×3×4 | 354 | 75.3 | 89.7 | 98.3 | 89.9 | 98.5 | 82.2 | 96.1 |
| UniFormerV2-L/14 336↑ | | 64×3×2 | 354 | 75.3 | **90.0** | **98.4** | **90.1** | **98.5** | **82.7** | 96.2 |
| UniFormerV2-L/14 336↑ | | 64×3×4 | 354 | 150.6 | **90.0** | **98.4** | **90.1** | **98.5** | **82.7** | **96.3** |
| UniFormerV2-L/14 (frozen) 336↑ | CLIP-400M | 8×1×3 | 51 | 4.7 | 86.7 | 93.4 | 87.4 | 97.7 | 79.6 | 94.6 |
| UniFormerV2-L/14 (frozen) 336↑ | CLIP-400M +K710 | 8×1×3 | 51 | 4.7 | 87.8 | 98.0 | 88.2 | 98.0 | 79.7 | 94.7 |
| UniFormerV2-L/14 (frozen) 336↑ | | 32×3×1 | 51 | 18.8 | 88.8 | 98.1 | 89.1 | 98.2 | 80.6 | 95.2 |
| UniFormerV2-L/14 (frozen) 336↑ | | 32×3×4 | 51 | 75.3 | 88.9 | 98.2 | 89.2 | 98.2 | 80.8 | 95.4 |

Table 16: **More results on Kinetics-400, 600 and 700.**

## C  MORE ABLATION STUDIES

We conduct more ablation studies based on CLIP-ViT-B/16 (Radford et al., 2021).

**Output token combination.** When only using global token for classification, the top-1 accuracy drops from 84.4% to 81.8% in Table 12. It shows that both local and global output tokens are essential for maintaining performance.

**Kinetics pretraining for Something-Something.** Different from the prior works (Li et al., 2022a; Fan et al., 2021), in Table 13, we find that extra Kinetics pretraining harms the representation inherited from CLIP, leading to lower performance.

**Query number.** In Table 14, we try to increase the query number. However, more queries lead to severe overfitting, thus the performance drops.

**Different modules.** In Table 15, we compare our local MHRA with popular temporal modules, including simple mean pooling (Wang et al., 2016), divided and joint space-time MHSA (Bertasius et al., 2021), temporal convolution (Tran et al., 2018), temporal shift (Lin et al., 2019) and temporal transformer (Sharir et al., 2021). All the modules are inserted before all the spatial MHSA, except that the 6-layer temporal transformer is added after the backbone. The results shows that our local MHRA beats the previous methods, achieving 2.0% to 22.6% higher top-1 accuracy. It demonstrate the effectiveness of our local MHRA for temporal modeling.

| Method | Pretrain | Frame×Crop×Clip | Param. (M) | FLOPs (T) | MiT Top-1 | MiT Top-5 |
|---|---|---|---|---|---|---|
| UniFormerV2-B/16 | CLIP-400M | 8×3×4 | 115 | 1.8 | 42.2 | 71.3 |
| UniFormerV2-B/16 | CLIP-400M | 32×3×4 | 115 | 7.2 | 42.2 | 71.5 |
| UniFormerV2-B/16 | CLIP-400M+K710 | 8×3×4 | 115 | 1.8 | **42.6** | 71.6 |
| UniFormerV2-B/16 | CLIP-400M+K710+K400 | 8×3×4 | 115 | 1.8 | **42.6** | **71.7** |
| UniFormerV2-B/16 | CLIP-400M+K710+K700 | 8×3×4 | 115 | 1.8 | 42.4 | 71.2 |
| UniFormerV2-L/14 | CLIP-400M | 8×3×4 | 354 | 8.0 | 46.2 | 76.0 |
| UniFormerV2-L/14 | CLIP-400M | 16×3×4 | 354 | 16.0 | 46.2 | **76.2** |
| UniFormerV2-L/14 | CLIP-400M | 32×3×4 | 354 | 32.0 | 46.4 | **76.2** |
| UniFormerV2-L/14 | CLIP-400M+K710 | 8×3×4 | 354 | 8.0 | 46.7 | **76.2** |
| UniFormerV2-L/14 | CLIP-400M+K710+K400 | 8×3×4 | 354 | 8.0 | **47.0** | 76.1 |
| UniFormerV2-L/14 336↑ | CLIP-400M | 8×3×4 | 354 | 18.8 | 47.2 | 76.5 |
| UniFormerV2-L/14 336↑ | CLIP-400M+K710 | 8×3×4 | 354 | 18.8 | 47.6 | 76.7 |
| UniFormerV2-L/14 336↑ | CLIP-400M+K710+K400 | 8×3×4 | 354 | 18.8 | **47.8** | **76.9** |

Table 17: **More results on Moments in Time V1.**

| Method | Frame×Crop×Clip | Param. (M) | FLOPs (T) | SSV1 Top-1 | SSV1 Top-5 | SSV2 Top-1 | SSV2 Top-5 |
|---|---|---|---|---|---|---|---|
| UniFormerV2-B/16 | 16×3×1 | 163 | 0.6 | 56.8 | 84.2 | 69.5 | 92.3 |
| UniFormerV2-B/16 | 16×3×2 | 163 | 1.1 | 57.2 | 84.3 | 69.7 | 92.5 |
| UniFormerV2-B/16 | 32×3×1 | 163 | 1.1 | 59.4 | **86.2** | 70.7 | **93.2** |
| UniFormerV2-B/16 | 32×3×2 | 163 | 2.2 | **59.5** | **86.2** | **71.0** | **93.2** |
| UniFormerV2-L/14 | 16×3×1 | 574 | 2.6 | 60.5 | 86.5 | 72.1 | 93.6 |
| UniFormerV2-L/14 | 16×3×2 | 574 | 5.2 | 60.9 | 86.8 | 72.2 | 93.7 |
| UniFormerV2-L/14 | 32×3×1 | 574 | 5.2 | 62.7 | 88.0 | 73.0 | **94.5** |
| UniFormerV2-L/14 | 32×3×2 | 574 | 10.3 | **62.9** | **88.3** | **73.1** | **94.5** |

Table 18: **More results on Something-Something.** All models are directly finetuned from CLIP.

| Dataset | Pretrain | Frame | 3×2 Top-1 | 3×2 Top-5 | 3×4 Top-1 | 3×4 Top-5 | 3×10 Top-1 | 3×10 Top-5 |
|---|---|---|---|---|---|---|---|---|
| ActivityNet | CLIP-400M+K400 | 8 | 92.8 | 99.0 | 92.8 | 99.1 | 93.0 | 99.1 |
| | CLIP-400M+K400 | 16 | 93.5 | 99.4 | 93.5 | 99.5 | 93.6 | 99.5 |
| | CLIP-400M+K710+K400 | 16 | 93.9 | 99.4 | 94.1 | 99.5 | 94.3 | 99.5 |
| | CLIP-400M+K710+K700 | 16 | 94.0 | 99.4 | 94.2 | 99.5 | 94.3 | **99.6** |
| | CLIP-400M+K710+K400 | 32 | 94.3 | **99.6** | 94.5 | **99.6** | **94.7** | 99.5 |
| HACS | CLIP-400M+K400 | 16 | 94.7 | 99.8 | 94.7 | 99.8 | 94.9 | **99.9** |
| | CLIP-400M+K710+K400 | 16 | 95.3 | 99.9 | 95.2 | 99.8 | **95.5** | 99.8 |
| | CLIP-400M+K710+K700 | 16 | 94.7 | 99.7 | 94.7 | 99.8 | 94.9 | 99.8 |
| | CLIP-400M+K710+K400 | 32 | 95.2 | 99.8 | 95.3 | 99.8 | 95.4 | 99.8 |

Table 19: **More results on ActivityNet and HACS.** All models are based on UniFormerV2-L/14.

# D    ADDITIONAL RESULTS

In Table 16, Table 17, Table 18 and Table 19, we give more results on the 8 video benchmarks, i.e., Kinetics-400/600/700, Moments in Time, Something-Something V1/V2, ActivityNet and HACS.

# E    MORE DISCUSSIONS

**Local UniBlock vs. ST-Adapter (Pan et al., 2022).** Our Local UniBlock is motivated by the style of UniForme r(Li et al., 2022a), i.e., we treat temporal depth-wise convolution as local temporal relation aggregator. Hence, like UniFormer, we introduce extra BatchNorm (Ioffe & Szegedy, 2015) before the first linear projection V(·). Alternatively, ST-adapter does not have this design, since it simply treats temporal depth-wise convolution as adaptation. With such motivation, it further introduces extra activation function for enhancing such adaptation, while our local UniBlock does not need it. In fact, we have also made comparisons in Table 9b. It shows that our local MHRA beats ST-Adapter (69.1% vs. 68.0%).

**Global UniBlock vs.  Perceiver (Jaegle et al., 2021), DETR (Carion et al., 2020) and Flamingo(Alayrac et al., 2022).** Our Glocal UniBlock is also motivated by the style of UniFormer (Li et al., 2022a).  But differently, to decrease the global computation in UniFormer, we change

self-attention MHRA as cross-attention MHRA in our UniFormerV2. Hence, our Global UniBlock consists of Dynamic Position Embedding (DPE), cross MHRA and FFN. On the contrary, none of those works belong to such an operation combination, without insight of UniFormer in video learning. In fact, these methods often use the standard cross-style transformer block including self MHRA, cross MHRA and FFN.

**Limitations.** In UniFormerV2, we propose the effective designs to arm pretrained ViT as spatiotemporal learners. Although its training is more efficient compared to non-trivial video backbones, its performance tends to depend on the scale of pretraining data, as shown in Table 8. Hence, it would be interesting to explore our UniFormerV2 on huge image foundation models pretrained by massive datasets, for further evaluating its scalability and generalization capacity.

## F  LABEL LIST OF KINETICS-710

To generate our Kinetics-710, we align labels in different Kinetics datasets by filtering symbols and replacing synonyms. The final label list is shown in Table20. Compared with Kinetics-700, there are 8 and 2 unique labels in Kinetics-400 and Kinetics-600 respectively. When finetuning the models pretrained on Kinetics-710, it is vital to load the pretrained weight of the classification layer, thus we map the weight according to the label list.

Table 20: **Labels of Kinetics-710.**

| Label | K4 | K6 | K7 | Label | K4 | K6 | K7 | Label | K4 | K6 | K7 |
|---|---|---|---|---|---|---|---|---|---|---|---|
| luge | ✗ | ✓ | ✓ | krumping | ✓ | ✓ | ✓ | skiing mono | ✓ | ✓ | ✓ |
| yoga | ✓ | ✓ | ✓ | slapping | ✓ | ✓ | ✓ | ski jumping | ✓ | ✓ | ✓ |
| vault | ✓ | ✗ | ✗ | decoupage | ✗ | ✗ | ✓ | driving car | ✓ | ✓ | ✓ |
| squat | ✓ | ✓ | ✓ | arresting | ✗ | ✗ | ✓ | tap dancing | ✓ | ✓ | ✓ |
| lunge | ✓ | ✓ | ✓ | surveying | ✗ | ✗ | ✓ | hockey stop | ✓ | ✓ | ✓ |
| zumba | ✓ | ✓ | ✓ | fly tying | ✗ | ✓ | ✓ | tobogganing | ✓ | ✓ | ✓ |
| situp | ✓ | ✓ | ✓ | capsizing | ✗ | ✓ | ✓ | cooking egg | ✓ | ✓ | ✓ |
| sewing | ✗ | ✓ | ✓ | tiptoeing | ✗ | ✓ | ✓ | slacklining | ✓ | ✓ | ✓ |
| cumbia | ✗ | ✓ | ✓ | using atm | ✗ | ✓ | ✓ | pushing car | ✓ | ✓ | ✓ |
| crying | ✓ | ✓ | ✓ | waking up | ✗ | ✓ | ✓ | ice skating | ✓ | ✓ | ✓ |
| dining | ✓ | ✓ | ✓ | fidgeting | ✗ | ✓ | ✓ | ice fishing | ✓ | ✓ | ✓ |
| digging | ✓ | ✗ | ✓ | tie dying | ✗ | ✓ | ✓ | celebrating | ✓ | ✓ | ✓ |
| chasing | ✗ | ✗ | ✓ | wrestling | ✓ | ✓ | ✓ | windsurfing | ✓ | ✓ | ✓ |
| sieving | ✗ | ✗ | ✓ | whistling | ✓ | ✓ | ✓ | riding mule | ✓ | ✓ | ✓ |
| staring | ✗ | ✓ | ✓ | high kick | ✓ | ✓ | ✓ | waxing legs | ✓ | ✓ | ✓ |
| karaoke | ✗ | ✓ | ✓ | abseiling | ✓ | ✓ | ✓ | deadlifting | ✓ | ✓ | ✓ |
| burping | ✗ | ✓ | ✓ | high jump | ✓ | ✓ | ✓ | bee keeping | ✓ | ✓ | ✓ |
| packing | ✗ | ✓ | ✓ | trapezing | ✓ | ✓ | ✓ | pumping gas | ✓ | ✓ | ✓ |
| licking | ✗ | ✓ | ✓ | skydiving | ✓ | ✓ | ✓ | tapping pen | ✓ | ✓ | ✓ |
| winking | ✗ | ✓ | ✓ | bandaging | ✓ | ✓ | ✓ | headbanging | ✓ | ✓ | ✓ |
| arguing | ✗ | ✓ | ✓ | side kick | ✓ | ✓ | ✓ | bookbinding | ✓ | ✓ | ✓ |
| ironing | ✓ | ✓ | ✓ | jetskiing | ✓ | ✓ | ✓ | flying kite | ✓ | ✓ | ✓ |
| drawing | ✓ | ✓ | ✓ | long jump | ✓ | ✓ | ✓ | fixing hair | ✓ | ✓ | ✓ |
| archery | ✓ | ✓ | ✓ | hopscotch | ✓ | ✓ | ✓ | egg hunting | ✓ | ✓ | ✓ |
| jogging | ✓ | ✓ | ✓ | dodgeball | ✓ | ✓ | ✓ | mowing lawn | ✓ | ✓ | ✓ |
| singing | ✓ | ✓ | ✓ | crocheting | ✗ | ✗ | ✓ | triple jump | ✓ | ✓ | ✓ |
| yawning | ✓ | ✓ | ✓ | ski ballet | ✗ | ✗ | ✓ | milking cow | ✓ | ✓ | ✓ |
| writing | ✓ | ✓ | ✓ | geocaching | ✗ | ✓ | ✓ | doing nails | ✓ | ✓ | ✓ |
| push up | ✓ | ✓ | ✓ | bulldozing | ✗ | ✓ | ✓ | dyeing hair | ✓ | ✓ | ✓ |
| tai chi | ✓ | ✓ | ✓ | cosplaying | ✗ | ✓ | ✓ | eating cake | ✓ | ✓ | ✓ |
| sailing | ✓ | ✓ | ✓ | spelunking | ✗ | ✓ | ✓ | paragliding | ✓ | ✓ | ✓ |
| welding | ✓ | ✓ | ✓ | jaywalking | ✗ | ✓ | ✓ | headbutting | ✓ | ✓ | ✓ |
| smoking | ✓ | ✓ | ✓ | head stand | ✗ | ✓ | ✓ | bobsledding | ✓ | ✓ | ✓ |
| parkour | ✓ | ✓ | ✓ | contorting | ✗ | ✓ | ✓ | kitesurfing | ✓ | ✓ | ✓ |
| texting | ✓ | ✓ | ✓ | plastering | ✓ | ✓ | ✓ | petting cat | ✓ | ✓ | ✓ |
| bowling | ✓ | ✓ | ✓ | bartending | ✓ | ✓ | ✓ | waxing back | ✓ | ✓ | ✓ |
| kissing | ✓ | ✓ | ✓ | beatboxing | ✓ | ✓ | ✓ | making slime | ✗ | ✗ | ✓ |
| busking | ✓ | ✓ | ✓ | applauding | ✓ | ✓ | ✓ | steering car | ✗ | ✗ | ✓ |
| gargling | ✓ | ✗ | ✓ | pole vault | ✓ | ✓ | ✓ | rolling eyes | ✗ | ✗ | ✓ |
| spraying | ✓ | ✗ | ✓ | barbequing | ✓ | ✓ | ✓ | moving child | ✗ | ✗ | ✓ |
| coughing | ✗ | ✗ | ✓ | snowkiting | ✓ | ✓ | ✓ | pouring milk | ✗ | ✗ | ✓ |
| saluting | ✗ | ✗ | ✓ | making tea | ✓ | ✓ | ✓ | grooming cat | ✗ | ✗ | ✓ |
| shouting | ✗ | ✗ | ✓ | auctioning | ✓ | ✓ | ✓ | doing sudoku | ✗ | ✗ | ✓ |
| sleeping | ✗ | ✓ | ✓ | snorkeling | ✓ | ✓ | ✓ | closing door | ✗ | ✗ | ✓ |
| smashing | ✗ | ✓ | ✓ | testifying | ✓ | ✓ | ✓ | pouring wine | ✗ | ✗ | ✓ |
| tackling | ✗ | ✓ | ✓ | high fiving | ✗ | ✗ | ✓ | cutting cake | ✗ | ✗ | ✓ |
| shopping | ✗ | ✓ | ✓ | moving baby | ✗ | ✗ | ✓ | milking goat | ✗ | ✗ | ✓ |
| pinching | ✗ | ✓ | ✓ | shoot dance | ✗ | ✗ | ✓ | playing oboe | ✗ | ✗ | ✓ |
| huddling | ✗ | ✓ | ✓ | pirouetting | ✗ | ✓ | ✓ | filling cake | ✗ | ✗ | ✓ |
| bottling | ✗ | ✓ | ✓ | coloring in | ✗ | ✓ | ✓ | sanding wood | ✗ | ✗ | ✓ |
| drooling | ✗ | ✓ | ✓ | sawing wood | ✗ | ✓ | ✓ | jumping sofa | ✗ | ✗ | ✓ |
| tickling | ✓ | ✓ | ✓ | calculating | ✗ | ✓ | ✓ | taking photo | ✗ | ✗ | ✓ |
| knitting | ✓ | ✓ | ✓ | waving hand | ✗ | ✓ | ✓ | silent disco | ✗ | ✗ | ✓ |
| unboxing | ✓ | ✓ | ✓ | watching tv | ✗ | ✓ | ✓ | ironing hair | ✗ | ✓ | ✓ |
| shot put | ✓ | ✓ | ✓ | calligraphy | ✗ | ✓ | ✓ | planing wood | ✗ | ✓ | ✓ |
| marching | ✓ | ✓ | ✓ | carving ice | ✗ | ✓ | ✓ | gold panning | ✗ | ✓ | ✓ |
| capoeira | ✓ | ✓ | ✓ | bodysurfing | ✗ | ✓ | ✓ | pillow fight | ✗ | ✓ | ✓ |
| pull ups | ✓ | ✓ | ✓ | lifting hat | ✗ | ✓ | ✓ | combing hair | ✗ | ✓ | ✓ |
| laughing | ✓ | ✓ | ✓ | bathing dog | ✗ | ✓ | ✓ | laying stone | ✗ | ✓ | ✓ |
| hurdling | ✓ | ✓ | ✓ | chewing gum | ✗ | ✓ | ✓ | photobombing | ✗ | ✓ | ✓ |
| sneezing | ✓ | ✓ | ✓ | parasailing | ✓ | ✓ | ✓ | playing lute | ✗ | ✓ | ✓ |
| clapping | ✓ | ✓ | ✓ | sipping cup | ✓ | ✓ | ✓ | land sailing | ✗ | ✓ | ✓ |

| Label | K4 | K6 | K7 | Label | K4 | K6 | K7 | Label | K4 | K6 | K7 |
|---|---|---|---|---|---|---|---|---|---|---|---|
| scrapbooking | × | ✓ | ✓ | washing feet | ✓ | ✓ | ✓ | ripping paper | ✓ | ✓ | ✓ |
| tasting wine | × | ✓ | ✓ | diving cliff | ✓ | ✓ | ✓ | crawling baby | ✓ | ✓ | ✓ |
| docking boat | × | ✓ | ✓ | golf putting | ✓ | ✓ | ✓ | cleaning pool | ✓ | ✓ | ✓ |
| photocopying | × | ✓ | ✓ | motorcycling | ✓ | ✓ | ✓ | brushing hair | ✓ | ✓ | ✓ |
| clam digging | × | ✓ | ✓ | breakdancing | ✓ | ✓ | ✓ | sanding floor | ✓ | ✓ | ✓ |
| ice swimming | × | ✓ | ✓ | drinking beer | ✓ | × | × | belly dancing | ✓ | ✓ | ✓ |
| roasting pig | × | ✓ | ✓ | swinging legs | ✓ | × | × | feeding goats | ✓ | ✓ | ✓ |
| pouring beer | × | ✓ | ✓ | bull fighting | × | ✓ | × | shaking hands | ✓ | ✓ | ✓ |
| smoking pipe | × | ✓ | ✓ | tossing salad | ✓ | × | ✓ | swing dancing | ✓ | ✓ | ✓ |
| lock picking | × | ✓ | ✓ | playing cards | ✓ | × | ✓ | carrying baby | ✓ | ✓ | ✓ |
| steer roping | × | ✓ | ✓ | slicing onion | × | × | ✓ | bending metal | ✓ | ✓ | ✓ |
| hugging baby | × | ✓ | ✓ | stacking dice | × | × | ✓ | playing poker | ✓ | ✓ | ✓ |
| embroidering | × | ✓ | ✓ | helmet diving | × | × | ✓ | grinding meat | ✓ | ✓ | ✓ |
| longboarding | × | ✓ | ✓ | dealing cards | × | × | ✓ | shining shoes | ✓ | ✓ | ✓ |
| laying tiles | × | ✓ | ✓ | treating wood | × | × | ✓ | folding paper | ✓ | ✓ | ✓ |
| playing gong | × | ✓ | ✓ | eating nachos | × | × | ✓ | blasting sand | ✓ | ✓ | ✓ |
| base jumping | × | ✓ | ✓ | being excited | × | × | ✓ | arm wrestling | ✓ | ✓ | ✓ |
| playing polo | × | ✓ | ✓ | vacuuming car | × | × | ✓ | rock climbing | ✓ | ✓ | ✓ |
| moon walking | × | ✓ | ✓ | petting horse | × | × | ✓ | catching fish | ✓ | ✓ | ✓ |
| opening door | × | ✓ | ✓ | stacking cups | × | × | ✓ | playing drums | ✓ | ✓ | ✓ |
| tasting food | ✓ | ✓ | ✓ | poaching eggs | × | × | ✓ | cracking neck | ✓ | ✓ | ✓ |
| shaving legs | ✓ | ✓ | ✓ | yarn spinning | × | ✓ | ✓ | tying necktie | ✓ | ✓ | ✓ |
| pumping fist | ✓ | ✓ | ✓ | card stacking | × | ✓ | ✓ | juggling fire | ✓ | ✓ | ✓ |
| making sushi | ✓ | ✓ | ✓ | rope pushdown | × | ✓ | ✓ | golf chipping | ✓ | ✓ | ✓ |
| snowmobiling | ✓ | ✓ | ✓ | smelling feet | × | ✓ | ✓ | javelin throw | ✓ | ✓ | ✓ |
| tasting beer | ✓ | ✓ | ✓ | card throwing | × | ✓ | ✓ | skateboarding | ✓ | ✓ | ✓ |
| golf driving | ✓ | ✓ | ✓ | playing darts | × | ✓ | ✓ | laying bricks | ✓ | ✓ | ✓ |
| waxing chest | ✓ | ✓ | ✓ | chopping meat | × | ✓ | ✓ | playing piano | ✓ | ✓ | ✓ |
| faceplanting | ✓ | ✓ | ✓ | making cheese | × | ✓ | ✓ | playing flute | ✓ | ✓ | ✓ |
| eating chips | ✓ | ✓ | ✓ | crossing eyes | × | ✓ | ✓ | salsa dancing | ✓ | ✓ | ✓ |
| playing harp | ✓ | ✓ | ✓ | cracking back | × | ✓ | ✓ | eating burger | ✓ | ✓ | ✓ |
| spinning poi | ✓ | ✓ | ✓ | building lego | × | ✓ | ✓ | skipping rope | ✓ | ✓ | ✓ |
| front raises | ✓ | ✓ | ✓ | using inhaler | × | ✓ | ✓ | climbing tree | ✓ | ✓ | ✓ |
| reading book | ✓ | ✓ | ✓ | jumping jacks | × | ✓ | ✓ | washing hands | ✓ | ✓ | ✓ |
| shaking head | ✓ | ✓ | ✓ | using puppets | × | ✓ | ✓ | playing chess | ✓ | ✓ | ✓ |
| snowboarding | ✓ | ✓ | ✓ | sucking lolly | × | ✓ | ✓ | tango dancing | ✓ | ✓ | ✓ |
| scuba diving | ✓ | ✓ | ✓ | cutting apple | × | ✓ | ✓ | using computer | ✓ | × | × |
| bending back | ✓ | ✓ | ✓ | lighting fire | × | ✓ | ✓ | cleaning floor | ✓ | × | × |
| drop kicking | ✓ | ✓ | ✓ | surfing water | ✓ | ✓ | ✓ | exercising arm | ✓ | × | × |
| using segway | ✓ | ✓ | ✓ | playing organ | ✓ | ✓ | ✓ | baby waking up | ✓ | × | ✓ |
| ice climbing | ✓ | ✓ | ✓ | hoverboarding | ✓ | ✓ | ✓ | waxing armpits | × | × | ✓ |
| tossing coin | ✓ | ✓ | ✓ | feeding birds | ✓ | ✓ | ✓ | mixing colours | × | × | ✓ |
| cheerleading | ✓ | ✓ | ✓ | blowing glass | ✓ | ✓ | ✓ | carving marble | × | × | ✓ |
| blowing nose | ✓ | ✓ | ✓ | building shed | ✓ | ✓ | ✓ | peeling banana | × | × | ✓ |
| pushing cart | ✓ | ✓ | ✓ | setting table | ✓ | ✓ | ✓ | breaking glass | × | × | ✓ |
| water skiing | ✓ | ✓ | ✓ | doing laundry | ✓ | ✓ | ✓ | laying decking | × | × | ✓ |
| making pizza | ✓ | ✓ | ✓ | braiding hair | ✓ | ✓ | ✓ | brushing floor | × | × | ✓ |
| punching bag | ✓ | ✓ | ✓ | mopping floor | ✓ | ✓ | ✓ | herding cattle | × | × | ✓ |
| feeding fish | ✓ | ✓ | ✓ | tying bow tie | ✓ | ✓ | ✓ | blending fruit | × | × | ✓ |
| riding camel | ✓ | ✓ | ✓ | cutting nails | ✓ | ✓ | ✓ | seasoning food | × | × | ✓ |
| shaving head | ✓ | ✓ | ✓ | skiing slalom | ✓ | ✓ | ✓ | checking watch | × | × | ✓ |
| throwing axe | ✓ | ✓ | ✓ | making a cake | ✓ | ✓ | ✓ | massaging neck | × | ✓ | ✓ |
| grooming dog | ✓ | ✓ | ✓ | chopping wood | ✓ | ✓ | ✓ | leatherworking | × | ✓ | ✓ |
| curling hair | ✓ | ✓ | ✓ | somersaulting | ✓ | ✓ | ✓ | acting in play | × | ✓ | ✓ |
| air drumming | ✓ | ✓ | ✓ | riding a bike | ✓ | ✓ | ✓ | chiseling wood | × | ✓ | ✓ |
| training dog | ✓ | ✓ | ✓ | surfing crowd | ✓ | ✓ | ✓ | square dancing | × | ✓ | ✓ |
| disc golfing | ✓ | ✓ | ✓ | holding snake | ✓ | ✓ | ✓ | sausage making | × | ✓ | ✓ |
| hula hooping | ✓ | ✓ | ✓ | water sliding | ✓ | ✓ | ✓ | using a wrench | × | ✓ | ✓ |
| washing hair | ✓ | ✓ | ✓ | playing cello | ✓ | ✓ | ✓ | weaving fabric | × | ✓ | ✓ |
| cartwheeling | ✓ | ✓ | ✓ | throwing ball | ✓ | ✓ | ✓ | breathing fire | × | ✓ | ✓ |
| changing oil | ✓ | ✓ | ✓ | eating hotdog | ✓ | ✓ | ✓ | rolling pastry | × | ✓ | ✓ |
| hammer throw | ✓ | ✓ | ✓ | robot dancing | ✓ | ✓ | ✓ | cutting orange | × | ✓ | ✓ |

| Label | K4 | K6 | K7 | Label | K4 | K6 | K7 | Label | K4 | K6 | K7 |
|---|---|---|---|---|---|---|---|---|---|---|---|
| needle felting | × | ✓ | ✓ | flipping bottle | × | × | ✓ | tagging graffiti | × | ✓ | ✓ |
| skipping stone | × | ✓ | ✓ | splashing water | × | × | ✓ | raising eyebrows | × | ✓ | ✓ |
| scrubbing face | × | ✓ | ✓ | carrying weight | × | × | ✓ | threading needle | × | ✓ | ✓ |
| flint knapping | × | ✓ | ✓ | spinning plates | × | × | ✓ | popping balloons | × | ✓ | ✓ |
| shuffling feet | × | ✓ | ✓ | fencing (sport) | × | ✓ | ✓ | cooking scallops | × | ✓ | ✓ |
| throwing knife | × | ✓ | ✓ | curling (sport) | × | ✓ | ✓ | backflip (human) | × | ✓ | ✓ |
| fixing bicycle | × | ✓ | ✓ | separating eggs | × | ✓ | ✓ | falling off bike | × | ✓ | ✓ |
| making bubbles | × | ✓ | ✓ | playing ocarina | × | ✓ | ✓ | playing scrabble | × | ✓ | ✓ |
| counting money | ✓ | ✓ | ✓ | playing netball | × | ✓ | ✓ | visiting the zoo | × | ✓ | ✓ |
| applying cream | ✓ | ✓ | ✓ | polishing metal | × | ✓ | ✓ | mosh pit dancing | × | ✓ | ✓ |
| blowing leaves | ✓ | ✓ | ✓ | jumping bicycle | × | ✓ | ✓ | shucking oysters | × | ✓ | ✓ |
| shoveling snow | ✓ | ✓ | ✓ | trimming shrubs | × | ✓ | ✓ | looking at phone | × | ✓ | ✓ |
| brush painting | ✓ | ✓ | ✓ | playing marbles | × | ✓ | ✓ | throwing tantrum | × | ✓ | ✓ |
| making the bed | ✓ | ✓ | ✓ | blowdrying hair | × | ✓ | ✓ | tying shoe laces | × | ✓ | ✓ |
| playing tennis | ✓ | ✓ | ✓ | dyeing eyebrows | × | ✓ | ✓ | dancing macarena | ✓ | ✓ | ✓ |
| playing violin | ✓ | ✓ | ✓ | laying concrete | × | ✓ | ✓ | playing bagpipes | ✓ | ✓ | ✓ |
| tapping guitar | ✓ | ✓ | ✓ | playing pinball | × | ✓ | ✓ | eating ice cream | ✓ | ✓ | ✓ |
| picking apples | ✓ | ✓ | ✓ | dumpster diving | × | ✓ | ✓ | playing monopoly | ✓ | ✓ | ✓ |
| doing aerobics | ✓ | ✓ | ✓ | putting on sari | × | ✓ | ✓ | flipping pancake | ✓ | ✓ | ✓ |
| drinking shots | ✓ | ✓ | ✓ | playing maracas | × | ✓ | ✓ | getting a tattoo | ✓ | ✓ | ✓ |
| bungee jumping | ✓ | ✓ | ✓ | delivering mail | × | ✓ | ✓ | building cabinet | ✓ | ✓ | ✓ |
| shearing sheep | ✓ | ✓ | ✓ | preparing salad | × | ✓ | ✓ | playing clarinet | ✓ | ✓ | ✓ |
| juggling balls | ✓ | ✓ | ✓ | vacuuming floor | × | ✓ | ✓ | eating spaghetti | ✓ | ✓ | ✓ |
| stretching arm | ✓ | ✓ | ✓ | chiseling stone | × | ✓ | ✓ | drumming fingers | ✓ | ✓ | ✓ |
| news anchoring | ✓ | ✓ | ✓ | breaking boards | × | ✓ | ✓ | eating doughnuts | ✓ | ✓ | ✓ |
| smoking hookah | ✓ | ✓ | ✓ | climbing ladder | ✓ | ✓ | ✓ | playing trombone | ✓ | ✓ | ✓ |
| massaging back | ✓ | ✓ | ✓ | hurling (sport) | ✓ | ✓ | ✓ | moving furniture | ✓ | ✓ | ✓ |
| weaving basket | ✓ | ✓ | ✓ | throwing discus | ✓ | ✓ | ✓ | contact juggling | ✓ | ✓ | ✓ |
| making snowman | ✓ | ✓ | ✓ | recording music | ✓ | ✓ | ✓ | playing recorder | ✓ | ✓ | ✓ |
| checking tires | ✓ | ✓ | ✓ | playing trumpet | ✓ | ✓ | ✓ | wrapping present | ✓ | ✓ | ✓ |
| planting trees | ✓ | ✓ | ✓ | sled dog racing | ✓ | ✓ | ✓ | hitting baseball | ✓ | ✓ | ✓ |
| spray painting | ✓ | ✓ | ✓ | stomping grapes | ✓ | ✓ | ✓ | playing kickball | ✓ | ✓ | ✓ |
| stretching leg | ✓ | ✓ | ✓ | carving pumpkin | ✓ | ✓ | ✓ | cleaning gutters | ✓ | ✓ | ✓ |
| clean and jerk | ✓ | ✓ | ✓ | unloading truck | ✓ | ✓ | ✓ | cleaning windows | ✓ | ✓ | ✓ |
| peeling apples | ✓ | ✓ | ✓ | watering plants | ✓ | ✓ | ✓ | peeling potatoes | ✓ | ✓ | ✓ |
| dancing ballet | ✓ | ✓ | ✓ | playing ukulele | ✓ | ✓ | ✓ | playing keyboard | ✓ | ✓ | ✓ |
| making jewelry | ✓ | ✓ | ✓ | cleaning toilet | ✓ | ✓ | ✓ | looking in mirror | × | × | ✓ |
| grooming horse | ✓ | ✓ | ✓ | folding napkins | ✓ | ✓ | ✓ | walking on stilts | × | × | ✓ |
| playing guitar | ✓ | ✓ | ✓ | playing cymbals | ✓ | ✓ | ✓ | playing billiards | × | × | ✓ |
| sword fighting | ✓ | ✓ | ✓ | riding unicycle | ✓ | ✓ | ✓ | curling eyelashes | × | × | ✓ |
| washing dishes | ✓ | ✓ | ✓ | playing cricket | ✓ | ✓ | ✓ | playing beer pong | × | ✓ | ✓ |
| roller skating | ✓ | ✓ | ✓ | climbing a rope | ✓ | ✓ | ✓ | directing traffic | × | ✓ | ✓ |
| massaging feet | ✓ | ✓ | ✓ | scrambling eggs | ✓ | ✓ | ✓ | twiddling fingers | × | ✓ | ✓ |
| cleaning shoes | ✓ | ✓ | ✓ | opening present | ✓ | ✓ | ✓ | marriage proposal | × | ✓ | ✓ |
| bench pressing | ✓ | ✓ | ✓ | folding clothes | ✓ | ✓ | ✓ | making horseshoes | × | ✓ | ✓ |
| riding scooter | ✓ | ✓ | ✓ | waiting in line | ✓ | ✓ | ✓ | cracking knuckles | × | ✓ | ✓ |
| sweeping floor | ✓ | ✓ | ✓ | finger snapping | ✓ | ✓ | ✓ | adjusting glasses | × | ✓ | ✓ |
| brushing teeth | ✓ | ✓ | ✓ | riding elephant | ✓ | ✓ | ✓ | tightrope walking | × | ✓ | ✓ |
| trimming trees | ✓ | ✓ | ✓ | waxing eyebrows | ✓ | ✓ | ✓ | playing laser tag | × | ✓ | ✓ |
| baking cookies | ✓ | ✓ | ✓ | shuffling cards | ✓ | ✓ | ✓ | installing carpet | × | ✓ | ✓ |
| massaging legs | ✓ | ✓ | ✓ | walking the dog | ✓ | ✓ | ✓ | lawn mower racing | × | ✓ | ✓ |
| crossing river | ✓ | ✓ | ✓ | driving tractor | ✓ | ✓ | ✓ | standing on hands | × | ✓ | ✓ |
| eating carrots | ✓ | ✓ | ✓ | strumming guitar | ✓ | × | × | playing pan pipes | × | ✓ | ✓ |
| taking a shower | ✓ | × | × | filling eyebrows | ✓ | × | ✓ | playing ping pong | × | ✓ | ✓ |
| cooking chicken | ✓ | × | ✓ | playing rounders | × | × | ✓ | falling off chair | × | ✓ | ✓ |
| shredding paper | ✓ | × | ✓ | squeezing orange | × | × | ✓ | playing blackjack | × | ✓ | ✓ |
| metal detecting | × | × | ✓ | making latte art | × | × | ✓ | mushroom foraging | × | ✓ | ✓ |
| lighting candle | × | × | ✓ | opening coconuts | × | × | ✓ | playing harmonica | ✓ | ✓ | ✓ |
| using megaphone | × | × | ✓ | playing checkers | × | × | ✓ | cutting pineapple | ✓ | ✓ | ✓ |
| playing piccolo | × | × | ✓ | sword swallowing | × | ✓ | ✓ | sharpening knives | ✓ | ✓ | ✓ |
| entering church | × | × | ✓ | playing dominoes | × | ✓ | ✓ | playing badminton | ✓ | ✓ | ✓ |
| playing mahjong | × | × | ✓ | putting on shoes | × | ✓ | ✓ | getting a haircut | ✓ | ✓ | ✓ |

| Label | K4 | K6 | K7 |
|---|---|---|---|
| playing saxophone | ✓ | ✓ | ✓ |
| making a sandwich | ✓ | ✓ | ✓ |
| playing xylophone | ✓ | ✓ | ✓ |
| reading newspaper | ✓ | ✓ | ✓ |
| jumping into pool | ✓ | ✓ | ✓ |
| arranging flowers | ✓ | ✓ | ✓ |
| frying vegetables | ✓ | ✓ | ✓ |
| sharpening pencil | ✓ | ✓ | ✓ |
| playing accordion | ✓ | ✓ | ✓ |
| eating watermelon | ✓ | ✓ | ✓ |
| jumpstyle dancing | ✓ | ✓ | ✓ |
| playing paintball | ✓ | ✓ | ✓ |
| playing nose flute | × | × | ✓ |
| getting a piercing | × | ✓ | ✓ |
| wading through mud | × | ✓ | ✓ |
| wood burning (art) | × | ✓ | ✓ |
| using circular saw | × | ✓ | ✓ |
| assembling bicycle | × | ✓ | ✓ |
| blowing bubble gum | × | ✓ | ✓ |
| repairing puncture | × | ✓ | ✓ |
| poking bellybutton | × | ✓ | ✓ |
| putting on mascara | × | ✓ | ✓ |
| throwing snowballs | × | ✓ | ✓ |
| riding snow blower | × | ✓ | ✓ |
| shining flashlight | × | ✓ | ✓ |
| using a microscope | × | ✓ | ✓ |
| kicking field goal | ✓ | ✓ | ✓ |
| playing ice hockey | ✓ | ✓ | ✓ |
| playing controller | ✓ | ✓ | ✓ |
| cutting watermelon | ✓ | ✓ | ✓ |
| dancing charleston | ✓ | ✓ | ✓ |
| hugging (not baby) | ✓ | ✓ | ✓ |
| springboard diving | ✓ | ✓ | ✓ |
| playing basketball | ✓ | ✓ | ✓ |
| dunking basketball | ✓ | ✓ | ✓ |
| playing volleyball | ✓ | ✓ | ✓ |
| playing didgeridoo | ✓ | ✓ | ✓ |
| inflating balloons | ✓ | ✓ | ✓ |
| extinguishing fire | ✓ | ✓ | ✓ |
| pushing wheelchair | ✓ | ✓ | ✓ |
| chopping vegetables | × | ✓ | × |
| pulling rope (game) | × | × | ✓ |
| picking blueberries | × | × | ✓ |
| playing road hockey | × | × | ✓ |
| uncorking champagne | × | × | ✓ |
| polishing furniture | × | × | ✓ |
| playing with trains | × | ✓ | ✓ |
| pushing wheelbarrow | × | ✓ | ✓ |
| shaping bread dough | × | ✓ | ✓ |
| alligator wrestling | × | ✓ | ✓ |
| building sandcastle | × | ✓ | ✓ |
| doing jigsaw puzzle | × | ✓ | ✓ |
| opening wine bottle | × | ✓ | ✓ |
| putting on eyeliner | × | ✓ | ✓ |
| passing soccer ball | × | ✓ | ✓ |
| playing rubiks cube | × | ✓ | ✓ |
| using a power drill | × | ✓ | ✓ |
| putting on lipstick | × | ✓ | ✓ |
| kicking soccer ball | ✓ | ✓ | ✓ |
| cooking on campfire | ✓ | ✓ | ✓ |
| gymnastics tumbling | ✓ | ✓ | ✓ |
| clay pottery making | ✓ | ✓ | ✓ |

| Label | K4 | K6 | K7 |
|---|---|---|---|
| swimming backstroke | ✓ | ✓ | ✓ |
| skiing crosscountry | ✓ | ✓ | ✓ |
| answering questions | ✓ | ✓ | ✓ |
| assembling computer | ✓ | ✓ | ✓ |
| sticking tongue out | ✓ | ✓ | ✓ |
| biking through snow | ✓ | ✓ | ✓ |
| playing bass guitar | ✓ | ✓ | ✓ |
| shooting basketball | ✓ | ✓ | ✓ |
| blowing out candles | ✓ | ✓ | ✓ |
| rock scissors paper | ✓ | ✓ | ✓ |
| riding mountain bike | ✓ | × | × |
| playing slot machine | × | × | ✓ |
| swimming with sharks | × | × | ✓ |
| playing shuffleboard | × | × | ✓ |
| using a paint roller | × | ✓ | ✓ |
| home roasting coffee | × | ✓ | ✓ |
| battle rope training | × | ✓ | ✓ |
| changing gear in car | × | ✓ | ✓ |
| swimming front crawl | × | ✓ | ✓ |
| wading through water | × | ✓ | ✓ |
| walking through snow | × | ✓ | ✓ |
| attending conference | × | ✓ | ✓ |
| casting fishing line | × | ✓ | ✓ |
| opening refrigerator | × | ✓ | ✓ |
| hand washing clothes | × | ✓ | ✓ |
| playing field hockey | × | ✓ | ✓ |
| juggling soccer ball | ✓ | ✓ | ✓ |
| dribbling basketball | ✓ | ✓ | ✓ |
| country line dancing | ✓ | ✓ | ✓ |
| canoeing or kayaking | ✓ | ✓ | ✓ |
| running on treadmill | ✓ | ✓ | ✓ |
| walking with crutches | × | × | ✓ |
| pulling espresso shot | × | × | ✓ |
| letting go of balloon | × | × | ✓ |
| being in zero gravity | × | × | ✓ |
| roasting marshmallows | × | ✓ | ✓ |
| using bagging machine | × | ✓ | ✓ |
| talking on cell phone | × | ✓ | ✓ |
| putting on foundation | × | ✓ | ✓ |
| using a sledge hammer | × | ✓ | ✓ |
| swinging baseball bat | × | ✓ | ✓ |
| making balloon shapes | × | ✓ | ✓ |
| dancing gangnam style | ✓ | ✓ | ✓ |
| cooking sausages | ✓ | ✓ | ✓ |
| snatch weight lifting | ✓ | ✓ | ✓ |
| swinging on something | ✓ | ✓ | ✓ |
| swimming with dolphins | × | × | ✓ |
| shooting off fireworks | × | × | ✓ |
| throwing water balloon | × | ✓ | ✓ |
| historical reenactment | × | ✓ | ✓ |
| swimming breast stroke | ✓ | ✓ | ✓ |
| bouncing on trampoline | ✓ | ✓ | ✓ |
| shooting goal (soccer) | ✓ | ✓ | ✓ |
| riding mechanical bull | ✓ | ✓ | ✓ |
| making paper aeroplanes | × | ✓ | ✓ |
| using remote controller | ✓ | ✓ | ✓ |
| massaging person's head | ✓ | ✓ | ✓ |
| gospel singing in church | × | ✓ | ✓ |
| punching person (boxing) | ✓ | ✓ | ✓ |
| petting animal (not cat) | ✓ | ✓ | ✓ |
| pretending to be a statue | × | × | ✓ |
| listening with headphones | × | × | ✓ |

| Label | K4 | K6 | K7 |
|---|---|---|---|
| putting wallpaper on wall | × | × | ✓ |
| playing american football | × | × | ✓ |
| carving wood with a knife | × | × | ✓ |
| bouncing on bouncy castle | × | ✓ | ✓ |
| putting in contact lenses | × | ✓ | ✓ |
| archaeological excavation | × | ✓ | ✓ |
| swimming butterfly stroke | ✓ | ✓ | ✓ |
| tying knot (not on a tie) | ✓ | ✓ | ✓ |
| person collecting garbage | ✓ | ✓ | ✓ |
| trimming or shaving beard | ✓ | ✓ | ✓ |
| giving or receiving award | ✓ | ✓ | ✓ |
| breading or breadcrumbing | ✓ | ✓ | ✓ |
| opening bottle (not wine) | ✓ | ✓ | ✓ |
| sign language interpreting | ✓ | ✓ | ✓ |
| mountain climber (exercise) | × | ✓ | ✓ |
| playing hand clapping games | × | ✓ | ✓ |
| presenting weather forecast | ✓ | ✓ | ✓ |
| bouncing ball (not juggling) | × | × | ✓ |
| changing wheel (not on bike) | ✓ | ✓ | ✓ |
| catching or throwing frisbee | ✓ | ✓ | ✓ |
| riding or walking with horse | ✓ | ✓ | ✓ |
| catching or throwing softball | ✓ | ✓ | ✓ |
| playing squash or racquetball | ✓ | ✓ | ✓ |
| decorating the christmas tree | ✓ | ✓ | ✓ |
| catching or throwing baseball | ✓ | ✓ | ✓ |
| exercising with an exercise ball | ✓ | ✓ | ✓ |
| passing American football (in game) | ✓ | ✓ | ✓ |
| passing American football (not in game) | ✓ | ✓ | ✓ |

