# OpenReview forum: "UniFormerV2: Spatiotemporal Learning by Arming Image ViTs with Video UniFormer"
_ICLR.cc/2023/Conference — Submitted to ICLR 2023_

### Official Review · Reviewer_GtFh · 2022-10-21

**Confidence:** 5
**Correctness:** 4
**Technical Novelty And Significance:** 2
**Empirical Novelty And Significance:** 3
**Recommendation:** 6

**Clarity, Quality, Novelty And Reproducibility:**

The paper is well-written and easy to follow. The idea of using pretrained image ViT to boost video transformers is novel

**Strength And Weaknesses:**

Pros:
1. The idea of arming rich well-pretrained image ViTs with efficient Uniformer design for video performance improvement is novel and useful.
2. The proposed method achieves state-of-the-art results on many popular video benchmarks.

Cons:
1. Lack of analysis of limitations of the proposed method
2. What is the video pretrain method used in Table 1? Is it supervised or self-supervised? And how many epochs does it take?
3. Why not use video pretrain in Table 4 for SSV2? Is there any special consideration?


**Summary Of The Paper:**

This paper proposes UniFormerV2, which could arm the readily available and well-pretrained image ViT with efficient Uniformer designs. Extensive experiments demonstrate the effectiveness of the proposed method.

**Summary Of The Review:**

Overall, this paper proposes an effective way of using well-pretrained image ViTs to facilitate the performance of video ViTs. The proposed method achieves state-of-the-art results on 8 popular video benchmarks.

**After rebuttal**: I carefully read the discussion between Reviwer ikef and the authors. I agree that some contributions are overclaimed since they are not entirely new and bear some similarities with other architectures. As mentioned by AC, this may need a re-writing of the manuscript and a new round of review.  But I think this paper gives us some good insights and practices on designing competitive video backbones. I lean to weekly accept this paper.

---

> ### Author Response · Authors · 2022-11-17
> **Response to Reviewer GtFh**
>
> Thanks for your positive comments! We provide our feedback as follows.
>
> -----
>
> **Q1: Lack of analysis of limitations of the proposed method.**
>
> **A1:** Thanks for your suggestion! We have added the limitations in Appendix E (Page 20). In this paper, we propose the effective designs to arm pretrained ViT as spatiotemporal learners. Although its training is more efficient compared to non-trivial video backbones, its performance tends to depend on the scale of pretraining data, as shown in Table 8. Hence, it would be interesting to explore our UniFormerV2 on huge image foundation models pretrained by massive datasets, for further evaluating its scalability and generalization capacity.
>
> -----
>
> **Q2: What is the video pretrain method used in Table 1? Is it supervised or self-supervised? And how many epochs does it take?**
>
> **A2:** We simply adopt supervised pretraining on Kinetics-710 for 55 epochs. All the training details can be found in Table 11 (Page 16). Note that, compared to other CLIP-based models, our UniFormerV2 uses the similar training epochs but with better accuracy, e.g., EVL-L achieves 87.7% on K400 with 53 finetune epochs, while our UniFormerV2 achieves 88.8% on K400 with 55 pretraining epochs and 5 finetune epochs.
>
> -----
>
> **Q3: Why not use video pretrain in Table 4 for SSV2? Is there any special consideration?**
>
> **A3:** Thanks for this comment. We would like to make further clarification.
>
> (1) This setting would definitely reduce the training cost on SSV2, without extra video pretraining epochs on K400.
>
> (2) CLIP-pretraining can produce robust visual features for SSV2, due to large-scale multi-modal learning. Hence, extra video post-pretraining is not necessary.  In fact, we made this comparison in Table 13. The setting of (CLIP pretraining+K400 post-pretraining+SSV2 finetuning) is slightly worse than our default setting of (CLIP pretraining+SSV2 finetuning).  The main reason is that, K400 is relatively small (0.24M videos), compared to 400M training pairs in CLIP. More importantly, there is a domain gap between scene-related K400 and temporal-related SSV2. In this case, such post-pretraining is not necessary for further boosting SSV2.

---

> > ### Comment · Reviewer_GtFh · 2022-12-06
> > **Response to Rebuttal**
> >
> > The rebuttal addressed all my concerns and I recommend accepting it.

---

### Official Review · Reviewer_DTkU · 2022-10-23

**Confidence:** 5
**Correctness:** 3
**Technical Novelty And Significance:** 3
**Empirical Novelty And Significance:** 3
**Recommendation:** 6

**Clarity, Quality, Novelty And Reproducibility:**

This paper clearly demonstrates its motivation and proposed method with sufficient details for reproducibility. As an extension work of UniFormer, this paper demonstrates a couple of novel designs.

**Strength And Weaknesses:**

Strength:

This paper is well-written. The discussion of new architecture design and empirical study are mostly well done. The motivation is well justified and the proposed algorithm is easy to follow.

The major contribution of this paper can be summarized as:

1. it decouple the previous local uniblock with one local MHRA temporal and one global MHRA spatial blocks, which not only reduce the local temporal redundancy, but also inherit the effective image pretraining of ViT architecture.

2. This work provides extensive empirical study of multi-stage fusion methods, pre training pipelines, training cost and intermediate feature locations of UniFormer. This provides insights of how each component works in the UniFormer.

3. This paper demonstrates strong performance in various video benchmarks, outperforming previous state-of-the-art performance. I believe the pre-trained models of this work are able to contribute to the community in various downstream video-based tasks.

From Figure one, it can be seen that the UniFormerV2 achieves good accuracy-FLOPs balance. Compared to other methods, UniFormerV2 achieves better performance with lower cost.

The implementation details and ablation set-up are well introduced, which provides much convenience for other researchers to re-implement the proposed algorithm in this paper.

Weakness & Questions:

Compared to the Local UniBlock in the UniFormerV1, the current version looks like a hybrid block of MHRA and MHSA. It would be interesting to see the performance of swapping local MHRA temporal with the temporal MHSA. In this case, the Local UniBlock will be like a divided Space-time attention introduced in the TimsFormer[1]

When leveraging the MHSA from image pretrained ViT, how to deal with the gap between ViT features and UniBlock features. The input to Global MHRA Spatial in UniFormer and the one to ViT MHSA should be very different.

Thanks authors for doing lots of experiments in various datasets. However, from the limited number of apple-to-apple comparisons, the improvement of UniFormerV2 is not significant.(for example compared to the MViTv2)

[1] Is Space-Time Attention All You Need for Video Understanding? ICML, 2021

**Summary Of The Paper:**

This paper proposes a new video transformer backbone, which is named UniFormerV2. Compared to the previous UniFormer, this paper improves the original local Multi-Head Relation Aggregator (MHRA) with one local MHRA temporal and one global MHRA spatial blocks. In which, the global MHRA spatial block can leverage the pretrained ViT model. Extensive experimental results demonstrate the new state-of-the-art performance in various video benchmarks.


**Summary Of The Review:**

Overall, I would vote weakly accept for this paper. The main reasons are: 1. The new design of UniFormer is well introduced and verified. 2. Extensive experimental results in 8 popular video benchmarks demonstrate its effectiveness and will potentially contribute to the community by releasing the pre-trained model.

---

> ### Author Response · Authors · 2022-11-17
> **Response to Reviewer DTkU**
>
> Thanks for your constructive comments. We provide our feedback as follows.
>
> -----
>
> **Q1: It would be interesting to see the performance of swapping local MHRA temporal with the temporal MHSA.**
>
> **A1:** In Table 9(b), we have replaced our local MHRA with temporal MHSA, and the performance decreases as expected (69.1% vs. 65.2%). To clearly compare different local operations, we only add local UniBlock or other local designs on ViT (Table 15, Page 18). The result shows that our local MHRA beats all the other local operations, which demonstrates its effectiveness.
>
> | Local Deisgn  | SSV2 |
> | :------------------------------- | :------: |
> | Mean Pooling | 45.1 |
> | Divided Space-Time MHSA | 63.4 |
> | Joint Space-Time MHSA  | 65.8 |
> | Temporal Convolution | 65.6 |
> | Temporal MHRA | 67.7 |
>
> -----
>
> **Q2: How to deal with the gap between ViT features and UniBlock features? The input to Global MHRA Spatial in UniFormer and the one to ViT MHSA should be very different.**
>
> **A2:** To minimize the gap, we explore several strategies as shown in the implementation details (Page 17). First，we zero-initialize some layers, e.g., the last point-wise convolutions in the local temporal MHRA. Therefore, at the beginning of training, the input and output will keep the same as the original ViT. Second, we use a relatively small learning rate (e.g., 1e-5) and propose K710 for post-pretraining. In this case, the model will slowly update the new parameters for temporal modeling.
>
> -----
>
> **Q3: From the limited number of apple-to-apple comparisons, the improvement of UniFormerV2 is not significant. (for example compared to the MViTv2)**
>
> **A3:** Thanks for this comment. We would like to make further clarification.
>
> (1) On K400, under ImageNet pretraining, the accuracy of our UniFormerV2-L/16 is competitive to MViTv2[1] (Top-1: 85.4% vs. 86.1%）but with less computation (TFLOPs: 12.2 vs. 42.2). It shows efficiency of our spatio-temporal modeling designs. Moreover, MViTv2 has to be learned from scratch without available pretraining. In this case, its training cost is also much larger than that of our UniFormerV2. For example, UniFormerV2-L ONLY needs around 50% video fine-tuning epochs of MViTv2-L on K400.
>
> (2) On K600 and K700 (Table2), our UniFormerV2-L significantly outperforms MViTv2-L, even when we use the frozen backbone (K600:  89.1% vs. 87.5%, K700: 80.6% vs. 79.4%), showing its effectiveness.
>
> (3) To give the apple-to-apple comparison, we have tried our best to list the SOTA models in terms of training epoch, computation and accuracy. From K400 (Table1) and SthSthV2 (Table4), we can see that, our UniFormerV2 often achieves the preferable accuracy-computation balance, with much less training cost.
>
> -----
>
> **References**
>
> [1] Li, Yanghao et al. “MViTv2: Improved Multiscale Vision Transformers for Classification and Detection.” CVPR (2022).

---

> > ### Comment · Reviewer_DTkU · 2022-12-06
> > **Response to Rebuttal**
> >
> > Thanks Authors for the hardworking of preparing this rebuttal, which has addressed most of my questions.
> >
> > After reading the feedback from all reviewers, I would keep my initial rating as weakly accept.
> >
> > Here is the main reasons behind this decision:
> >
> > 1. The proposed model achieves top performance in various benchmarks, which I believe will contribute many downstream applications in the community.
> >
> > 2. The proposed method is relatively efficient, which can be used for large-scaled training.
> >
> > However, I also agree with other reviewers regarding the following drawbacks:
> >
> > 1. The claimed improvement is minor, and the number of apple-to-apple comparison is less than expectation.
> >
> > 2. The novelty is limited.

---

> > > ### Author Response · Authors · 2022-12-06
> > > **Updated responses**
> > >
> > > Thank you for your feedback!
> > >
> > > Except for accuracy, we think training cost is also vital when comparing.  We recognize that MViT and UniFormer achieve better performance than UniFormerV2 with ImageNet-21K pretraining, but these elaborate architectures require tremendous training costs.  When arming with free CLIP-ViT, our UniFormerV2 simply achieves better performance.
> > >
> > > Besides, more apple-to-apple comparisons can be found in Table 8 and Table 15.  Our UniFormerV2 brings significant performance gains.  For example, compared with TimeSformer, it achieves 2.9% and 8.0% higher top-1 accuracy on K400 and SSV2, respectively.

---

### Official Review · Reviewer_ikef · 2022-10-23

**Confidence:** 4
**Correctness:** 3
**Technical Novelty And Significance:** 2
**Empirical Novelty And Significance:** 2
**Recommendation:** 3

**Clarity, Quality, Novelty And Reproducibility:**

I think clarity could have been improved if the authors chose to present this work as a CLIP adaptation for video recognition emphasizing the novel parts of their method.

**Details Of Ethics Concerns:**

No ethical concerns

**Strength And Weaknesses:**

Strengths:
+ CLIP adaptation methods is a reasonable direction for improving video recognition although the best models are really huge.
+ The architecture is evaluated on a large number of benchmarks where the method is shown to be among the most accurate ones.

Weaknesses:
- The method proposed in this paper is primarily an adaptation of CLIP for video recognition and it should have been presented as such. On the contrary the authors have tried to present Uniformer-V2 as a "generic paradigm to build a powerful family of video networks".
- The novelty of the proposed local and global uni-blocks seems thin: The local uni-block resembles a temporal depthwise convolution (can the authors make this clear as in the uniformer paper?) followed by a spatial transformer block as in the image encoder of CLIP. Hence, this block is basically a simple temporal adapter that is put on top of CLIP's ViT (something similar I believe has been proposed in Pan 2022). The global block resembles a perceiver block (ICML 2021) and/or DETR decoder (ECCV 2020) and in a similar fashion has been used in Flamingo (Neurips 2022). I think the authors need to discuss the similarities with these works.
- The experimental results are the paper's strongest point but they are not so impressive as they look from a first glance: (1) by using CLIP, there's a huge advantage over all other methods (which dont use CLIP), but that comes to no surprise. (2) When CLIP is not used, as in Table 1 (upper part), Uniformer-V2 is not better than Uniformer (which is trained on IN-1K). (3) When CLIP is used, the best results are reported by finetuning on their K710 (can you please clarify why this is needed? and explain in relation to table 9e) (4) On Something-Something V2, Uniformer outperforms Uniformer-V2 for the same comp. budget.

**After rebuttal:**
Unfortunately, the rebuttal does not address my concerns as also explained in my responses to the authors' responses. I still believe that the proposed blocks (local and global) must be described in a more clear manner to fully understand the differences to previous work. As it stands the proposed contributions seem thin. Moreover, as demonstrated in my review, the results are not so significant as claimed. For example on SS-V2, their method is outperformed by previous work and on K400 to surpass EVL their method needs additional pre-training.  Overall the paper shows that if CLIP is used one can achieve very high results on video recognition which is not a surprising result at all. Hence, I would like to keep my original score.


**Summary Of The Paper:**

This paper describes a new improved architecture for video recognition based on Uniformer (ICLR 2022), dubbed Uniformer-V2. To this end some of the techniques proposed in Uniformer are re-used and extended to create a video architecture that can benefit from existing pre-trained ViT models (CLIP in particular). The architecture is  evaluated on a number of video/action recognition benchmarks where the method is shown to be among the best reported method.

**Summary Of The Review:**

The paper has some merits but it's confusingly written. Moreover, the contributions in my opinion are not so strong as claimed. Some of the experimental results appear strong, but a closer look reveals that, in many cases, they are not so impressive as might have been thought from a first glance.

---

> ### Author Response · Authors · 2022-11-17
> **Response to Reviewer ikef (Part1)**
>
> Thanks for your constructive comments. We provide our feedbacks as follows.
>
> ----
>
> **Q1: The method proposed in this paper is primarily an adaptation of CLIP, but the authors have tried to present Uniformer-V2 as a "generic paradigm to build a powerful family of video networks".**
>
> **A1:** We clarify the difference between our work and CLIP adaptation.
>
> (1) Our UniFormerV2 aims to make good use of image-pretrained ViTs in multiple settings (not only CLIP) with efficient UniFormer-style designs. Hence, we show its effectiveness on different pretraining models and datasets. As shown in Table 8, we arm a large number of pretrained ViTs as our UniFormer V2. All of them surpass TimeSformer. This clearly supports our claim of building a powerful family of video networks by UniFormer V2.
>
> (2) Current CLIP adaptation approaches [1] often freeze the CLIP backbone and only finetune extra-introduced parameters. However, such a style is limited to capturing temporal-related dynamics (e.g., SthSth), due to the lack of end-to-end training on effective spatial-temporal architecture. On the contrary, our UniFormer V2 can achieve this goal. Like other popular video transformers (e.g., TimeSformer[2] and ViViT[3]), we design temporal modeling based on image-pretrained ViTs, and perform end-to-end training for effective local and global video relation learning.  As shown in Table 4, UniFormerV2 significantly outperforms the CLIP adaptation method (e.g., EVL[1]).
> In summary, we aim at providing good practices for arming image ViTs as spatiotemporal learners. UniFormerV2 is a generic and efficient paradigm for vanilla ViTs in both base and large model scales. We hope it can be a practical technique to explore and reuse the current image foundation models for video representation learning.
>
> ----
>
> **Q2: The authors need to discuss similarities with the previous works, w.r.t. the design of local and global uni-blocks.**
>
> **A2:** Thanks for your suggestions. We clarify the design differences here and have added these discussions in Appendix E (Page 19).
>
> (1) Local UniBlock vs. ST-Adapter[4]:
>
> Note that, our Local UniBlock is motivated by the style of UniFormer[5], i.e., we treat temporal depth-wise convolution as local temporal relation aggregator. Hence, like UniFormer, we introduce extra BatchNorm before the first linear projection V(·).  Alternatively, ST-adapter[4] does not have this design, since it simply treats temporal depth-wise convolution as adaptation. With such motivation, it further introduces an extra activation function (e.g., GELU) for enhancing such adaptation, while our local UniBlock does not need it. In fact, we have also made comparisons in Table 9(b). It shows that our local MHRA beats ST-Adapter (69.1% vs. 68.0%).
>
> (2) Global UniBlock vs. Perceiver[6], DETR[7] and Flamingo[8]:
>
> Again, our Glocal UniBlock is motivated by the style of UniFormer[5]. But differently,  to decrease the global computation in UniFormer, we change self-attention MHRA to cross-attention MHRA in our UniFormer V2. Hence, our Global UniBlock consists of Dynamic Position Embedding (DPE), cross MHRA and FFN. On the contrary, none of [6][7][8] belong to such an operation combination, without the insight of UniFormer in video learning. In fact, these methods often use the standard cross-style transformer block including self MHRA, cross MHRA and FFN.
>
> -----
>
> **Q3: By using CLIP, there's a huge advantage over all other methods (which dont use CLIP), but that comes to no surprise.**
>
> **A3:**  Thanks for this comment. We would like to make further clarification.
>
> (1) CLIP training on 400M image-text data does show its effectiveness on scene-related K400. But this is not dominant. Video network design is also important. For example, both CoCa[9] (88.9%) and MTV-H[10] (89.9%) do not use CLIP-pretraining, while they outperform X-CLIP-L[11] (using CLIP, 87.7%) with a notable margin. Our UniFormer V2 achieves the best, since we take advantage of both image pretraining and video model design, by arming well-pretrained ViTs with efficient UniFormer style，as shown in Figure 1 and Table 1.
>
> (2) CLIP pretraining does not work well on temporal-related Sth-Sth. As shown in Table 4, MTV-B (68.5%) does not use CLIP pretraining, while it outperforms EVL-L (66.7%, using CLIP). Once again, our UniFormer V2 achieves the best, since we also introduce effective spatiotemporal learning modules besides CLIP-pretraining.

---

> > ### Author Response · Authors · 2022-11-17
> > **Response to Reviewer ikef (Part2)**
> >
> > **Q4: When CLIP is not used, as in Table 1 (upper part), Uniformer-V2 is not better than Uniformer (which is trained on IN-1K).**
> >
> > **A4:** Note that, UniFormerV2 ONLY uses 50% finetuning epochs to achieve such competitive performance of UniFormer (Epoch: 55 vs 110, Top-5: 96.2% vs 95.4%) in this setting. It is also an important reason why we design UniFormerV2. As mentioned in our introduction, UniFormer has to be learned from scratch, leading to the expensive training cost and being hard to scale up for larger backbones. Alternatively,  our UniFormer V2 can efficiently alleviate such challenges, based on various open-sourced ViTs that are readily available and well-pretrained with rich image supervision.
> >
> > -----
> >
> > **Q5: When CLIP is used, the best results are reported by finetuning on their K710 (can you please clarify why this is needed? and explain in relation to table 9e)**
> >
> > **A5:** Without K710 pretraining, our UniFormerV2 already surpasses other methods, as shown in Figure 1. We further build up K710 for the following advantages, based on Table 9e.
> >
> > (1) Better generality:  K710-Pretraining (Row4) clearly improves the downstream performance on Kinetics-400, 600 and 700, compared with None-Pretraining (Row1 of Table 9e). It is also better than K400/600/700 co-training, especially on K700 (Row3 vs. Row2). It reveals that our compact K710 pretraining generalizes discriminative power of spatiotemporal representation. Such a conclusion can also be found in Table 16.
> >
> > (2) Less training cost: As we claimed in Appendix A (Page 17), our K710 pretraining requires less training cost. In Row1 of Table 9e, we finetune our model individually on Kinetics-400, 600 and 700. Due to the lack of pretraining, finetuning requires 55 epochs on each dataset for convergence. The total data size of these three datasets is 0.24 + 0.366 + 0.529 ≈ 1.14M. Hence, the rough training cost is 55x1.14=62.7. On the contrary, in Row4, we use K710 pretraining which also requires 55 epochs. The data size of K710 is 0.66M. After such pretraining, we ONLY finetune the model for 5 epochs on each dataset. Hence, the rough training cost is 55x0.66+5x1.14=42. Hence, we can save (1-42/62.7)*100%=33% training cost by our K710 pertaining.
> >
> > -----
> >
> > **Q6: On Something-Something V2, Uniformer outperforms Uniformer-V2 for the same comp. budget.**
> >
> > **A6:** The accuracy of UniFormerV2 is actually competitive to UniFormer[5] under the same setting (Base, 32×3×1): lower Top1 (70.7% vs. 71.2%) but higher Top5 (93.2% vs. 92.8%). However, the training cost of UniFormerV2 is way much smaller than UniFormer. As shown in Table 4, UniFormer-B needs 110 pretraining epochs on K400 and then 50 finetuning epochs on SSV2. In contrast, our UniFormerV2-B ONLY needs 22 finetuning epochs on SSV2. This is an important motivation to design UniFormerV2 for effective and efficient video learning. It also allows UniFormerV2 to scale up on the large backbone, which would be expensive for training UniFormer, due to learning from scratch.
> >
> > -----
> >
> > **References**
> >
> > [1] Lin, Ziyi et al. “Frozen CLIP Models are Efficient Video Learners.” ECCV (2022).
> >
> > [2] Bertasius, Gedas et al. “Is Space-Time Attention All You Need for Video Understanding?” ICML (2021).
> >
> > [3] Arnab, Anurag et al. “ViViT: A Video Vision Transformer.” ICCV (2021).
> >
> > [4] Pan, Junting et al. “Parameter-Efﬁcient Image-to-Video Transfer Learning.” NIPS (2022).
> >
> > [5] Li, Kunchang et al. “UniFormer: Unified Transformer for Efficient Spatiotemporal Representation Learning.” ICLR (2022).
> >
> > [6] Jaegle, Andrew et al. “Perceiver: General Perception with Iterative Attention.” ICML (2021).
> >
> > [7] Carion, Nicolas et al. “End-to-End Object Detection with Transformers.” ECCV (2020).
> >
> > [8] Alayrac, Jean-Baptiste et al. “Flamingo: a Visual Language Model for Few-Shot Learning.” ArXiv (2022).
> >
> > [9] Yu, Jiahui et al. “CoCa: Contrastive Captioners are Image-Text Foundation Models.” TMLR (2022).
> >
> > [10] Yan, Shen et al. “Multiview Transformers for Video Recognition.” CVPR (2022).
> >
> > [11] Ni, Bolin et al. “Expanding Language-Image Pretrained Models for General Video Recognition.” ECCV (2022).

---

> > > ### Comment · Reviewer_ikef · 2022-12-05
> > > **response to authors' responses**
> > >
> > > A4: I believe that the claimed advantage (55 training epochs vs 22) is quite minor. Importantly, it is hard to prove that this does not come from CLIP initialisation but from Uniformer-V2 design. Finally, Uniformer-V1 outperforms V2 on SS-V2.
> > >
> > > A5: Thank you for pointing to Fig. 1. Actually it shows that without K-710 pre-training Uniformer-V2 performs literally the same as EVL. I think this for clarity purposes should be included in the tables
> > >
> > > A6: Please see my response to A4.

---

> > > > ### Author Response · Authors · 2022-12-06
> > > > **Updated responses**
> > > >
> > > > **Reviewer A5: Thank you for pointing to Fig. 1. Actually it shows that without K-710 pre-training Uniformer-V2 performs literally the same as EVL. I think this for clarity purposes should be included in the tables.**
> > > >
> > > > Note that, EVL uses the multi-crop and single-clip testing strategy (e.g., 32x3x1 or 32x3) to boost its performance. On the contrary, we follow most existing approaches [1,2,3] with multi-crop and multi-clip testing strategy (e.g., 8x3x4 or 8x12). To make the state-of-the-art comparison, we show EVL with its best 32x3 setting in Fig. 1.
> > > >
> > > > In fact, as we show in Table16, under the **SAME** testing strategy, our UniFormerV2 without K710 pretraining is clearly **BETTER** than EVL on K400:
> > > >
> > > >     (1) B/16 with 8x3 testing: 84.0% vs. 82.9%
> > > >     (2) L/14 with 8x3 testing: 87.3% vs. 86.3%
> > > >
> > > > Besides, without K710 pretraining, our UniFormerV2 significantly outperforms EVL on SSV2:
> > > >
> > > >     (1) B/16 with 16x3 testing: 69.5% vs. 61.0%
> > > >     (2) L/14 with 32x3 testing: 73.1% vs. 66.7%
> > > >
> > > > ---
> > > >
> > > > **References**
> > > >
> > > > [1] Liu, Ze et al. “Video Swin Transformer.” CVPR (2021).
> > > >
> > > > [2] Arnab, Anurag et al. “ViViT: A Video Vision Transformer.” ICCV (2021).
> > > >
> > > > [3] Yan, Shen et al. “Multiview Transformers for Video Recognition.” CVPR (2022).

---

> > > > > ### Comment · Reviewer_ikef · 2022-12-07
> > > > > **further and final comment**
> > > > >
> > > > > I think it's clear from Fig. 1 that EVL and Uniformer-V2 when trained on the same dataset, and for the same computational cost, result literally in the same accuracy.

---

> > > > > > ### Author Response · Authors · 2022-12-07
> > > > > > **Updated response**
> > > > > >
> > > > > > Thanks again for your comment.
> > > > > >
> > > > > > We have claimed that in Figure 1, EVL achieves 'similar performance' because of the different testing strategy (more input frames but one sampling clip). Under the same testing strategy, UniFormerV2 obtains significant improvements.
> > > > > >
> > > > > > For a fair comparison, we follow most existing approaches to draw Figure 1 with a multi-clip testing strategy. Since EVL does not report the multi-clip testing results, we simply copy the single-clip testing results from its paper.

---

> > ### Comment · Reviewer_ikef · 2022-12-05
> > **responses to authors' response**
> >
> > A1: All of your main and best results are based on CLIP. Hence your method in my opinion falls into the category of CLIP adaptation methods.
> >
> > A2: I am sorry but I was not able to follow the differences you have highlighted in your response. For example, I still believe that the global block is very similar to perceiver.
> >
> > A3: You're right that CLIP pre-training might not be necessary for SS-V2. But on this dataset Uniformer-V1 beats Uniformer-V2.

---

> > > ### Author Response · Authors · 2022-12-06
> > > **Updated responses**
> > >
> > > **Reviewer A1: All of your main and best results are based on CLIP. Hence your method in my opinion falls into the category of CLIP adaptation methods.**
> > >
> > > We argue that CLIP adaptation refers to those methods freezing the CLIP backbone and only fine-tuning extra-introduced parameters. This is different from our goal. We aim at providing a practical end-to-end learning paradigm to build a video ViT, like the motivation of TimeSformer. Hence, we arm the image-pretrained ViTs with efficient UniFormer designs, and **FULLY** fine-tune all the layers in an **END-TO-END** manner for spatiotemporal representation learning.
> > >
> > > ---
> > >
> > > **Reviewer A2: I am sorry but I was not able to follow the differences you have highlighted in your response. For example, I still believe that the global block is very similar to perceiver.**
> > >
> > > (1) The global block in our UniFormerV2 has Dynamic Position Embedding (DPE) for video modeling. No such operation in Perceiver.
> > >
> > > (2) Perceiver uses more than one query (called latent array) to encode the latent representation. Hence, it also introduces extra latent transformers to learn the relation between queries. Differently, our global block aims at summarizing a global video representation. Hence, we **ONLY** use one query, and **DO NOT** need extra latent transformers.
> > >
> > > (3) Perceiver uses the **SAME** input (key/value, also called byte array) in all the cross-attention layers. In contrast, we use **DIFFERENT** inputs for different cross-attention layers, in order to fuse different-level video representations for video classification.
> > >
> > > ---
> > >
> > > **Reviewer A3 & A4 & A6: You're right that CLIP pre-training might not be necessary for SS-V2. But on this dataset Uniformer-V1 beats Uniformer-V2. I believe that the claimed advantage (55 training epochs vs 22) is quite minor. Importantly, it is hard to prove that this does not come from CLIP initialisation but from Uniformer-V2 design. Finally, Uniformer-V1 outperforms V2 on SS-V2.**
> > >
> > > As we claimed in the previous A6, UniFormerV2 is actually competitive with UniFormer under the same setting (Base, 32×3×1): lower Top1 (70.7% vs. 71.2%) but higher Top5 (93.2% vs. 92.8%). More importantly, the video training cost of UniFormer is actually much larger than UniFormerV2. As shown in Table 4, UniFormer-B has to use **110** pretraining epochs on K400 and then **50** finetuning epochs on SSV2. In contrast, our UniFormerV2-B **ONLY** needs **22** finetuning epochs on SSV2. The tremendous training cost limits scaling up UniFormer, while UniFormerV2 can efficiently make good use of large pretrained models.
> > >
> > > As shown in Table 9(a), without UniFormerV2 design, the top-1 accuracy drops 1.3% on Kinetics and 24.5% on SSV2. Table 8 also reflects that, compared with TimeSformer, our ImgaeNet-21K pretrained UniFormerV2 achieves 2.9% and 8.0% performance gain on K400 and SSV2 respectively. These results clearly show that the UniFormerV2 design brings a strong ability for spatiotemporal modeling.

---

> > > > ### Comment · Reviewer_ikef · 2022-12-07
> > > > **further and final response**
> > > >
> > > > As far as I know adaptation does not necessarily mean having the backbone frozen. End-to-end fine-tuning, LayerNom fine-tuning, adapters, to my knowledge, are all adaptation methods. The fact that you're fine-tuning everything is actually a disadvantage of your approach as a new set of weights must be stored.
> > > >
> > > > Thanks for the clarification about the differences with perceiver.  In my opinion, you block is still similar to the one proposed by perceiver and detr where "external" model parameters are injected into the backbone via cross attention (e.g. the fact that you're using one query doesn't constitute such an important difference). It would have been better if the differences were properly discussed rather than this being presented as something entirely new.
> > > >
> > > > I don't get why CLIP pre-training doesn't count at all when you make your calculations but Uniformer-V1 pre-training on K400 does count. Anyone can download Uniformer-V1 K400 pre-trained model and fine-tune it on any other video dataset.
> > > > Furthermore for your results on K400,  you do pre-train Uniformer-V2 which contradicts your point about pre-training.

---

> > > > > ### Author Response · Authors · 2022-12-07
> > > > > **Updated responses**
> > > > >
> > > > > Thanks for your patient responses. It's hard for us to reach an agreement. We finally want to reclaim why we consider the training cost of UniFormerV1.
> > > > >
> > > > > As elaborate architecture, UniFormerV1 requires to be trained from scratch, which limits its ability to scale up. It's difficult for most researchers to train a large model. In contrast, great efforts have been made to build powerful vanilla ViTs, e.g., CLIP and BeiT. These vanilla ViTs are open-sourced and will be stronger in the future. We think it will be popular to utilize strong pretrained ViTs in the video domain, just like widely-used ImageNet pretrained ResNet in the past few years.
> > > > >
> > > > > We hope UniFormerV2 can provide good practices for spatiotemporal modeling, and motivate researchers to make good use of the pretrained ViTs efficiently and effectively.

---

### Official Review · Reviewer_AXo7 · 2022-10-25

**Confidence:** 4
**Correctness:** 4
**Technical Novelty And Significance:** 2
**Empirical Novelty And Significance:** 3
**Recommendation:** 6

**Clarity, Quality, Novelty And Reproducibility:**

The paper is overall well organized and well written. The description of UniformerV2 is clear and easy to follow, and the model should be able to reproduced. The originality of the work is not that significant because many of the modules are not new (e.g., the affinity operation in local temporal aggregation, the learnable-query-based attention in global temporal attention, and the multi-scale fusion designs). Nevertheless, the contribution of this paper is that it provides detailed ablation to these design choices and verify the effectiveness of these modules.

**Strength And Weaknesses:**

Strengths
1. The paper presents strong experimental results on multiple video recognition datasets, including those requires stronger capacity on temporal modeling (e.g., Something-something) and long-range modeling (e.g., ActivityNet). The paper also provides extensive ablation study to present the impact of each component of the model and different design choices. These design choices are good practices for video understanding research and could be valuable for the community.


Minor questions:
1. How is the 3D convolution for patch embedding initialized? Is it inflated from patch embedding weights in the ViTs?
2. The four multi-stage fusion strategies share similar results in the ablation study. Which one would the author prefer to use in this case? Other than recognition accuracy, how about the comparison of computational cost / running time among them?

**Summary Of The Paper:**

This paper proposes a new video Transformer architecture, termed UniformerV2, that extends the pre-trained ViTs (on images) to the video action recognition tasks. Specifically, UniformerV2 introduces local temporal aggregation, global temporal modeling and multi-stage fusion to enhance the original image-based ViTs to capture temporal information. Although some of the modules are inspired by the design in Uniformer, the paper proposes better and more efficient design choices for local / global temporal aggregation, and presents extensive ablation studies to verify their effectiveness. The paper reports experiment results on multiple video recognition benchmarks and UniformerV2 achieves state-of-the-art results.

**Summary Of The Review:**

This paper proposes a good design choice for extending pretrained ViTs to video recognition task. The proposed method achieves superior results on multiple video recognition benchmarks and the effectiveness of the model is verified by extensive ablation study. In summary, I suggest "accept" to this paper.

*After reviewers' discussion*: I agree with the other reviewers that the novelty of the proposed local & global block is not as significant as claimed in the paper. However, I do feel that this paper provides some good practice and insights for making use of pre-trained image-based ViTs for spatiotemporal modeling in video recognition. Although the individual modules are not novel, the whole system presented in this paper still differs from prior work on extending image ViTs to video modeling (e.g., ST-Adapter, X-CLIP) and achieves better results (or comparable results with better efficiency). By reconsidering the strength and weakness of this paper, I'd update my rating to "weak accept".

---

> ### Author Response · Authors · 2022-11-17
> **Response to Reviewer AXo7**
>
> Thanks for your positive comments! We also hope UniFormerV2 can provide good practices for spatiotemporal modeling, and motivate researchers to make good use of the pretrained ViTs. All the code and models will be available afterward.
>
> ------
>
> **Q1: How is the 3D convolution for patch embedding initialized? Is it inflated from patch embedding weights in the ViTs?**
>
> **A1:** Yes, we inflate the patch embedding weights from the image ViTs. Specifically, we pad zero along the temporal dimension, meaning only the middle kernel directly loads the pretrained weights while the remaining are all zero. In our experiments, this initialization works better than alternatives. For Sth-Sth V2, zero padding achieves 1.1% higher performance than averaged weights (69.5% vs. 68.4%).
>
> ------
> **Q2: The four multi-stage fusion strategies share similar results in the ablation study. Which one would the author prefer to use in this case? Other than recognition accuracy, how about the comparison of computational cost / running time among them?**
>
> **A2:** We prefer sequential fusion for its structural simplicity. Moreover, since we use one query token for cross-attention in each global UniBlock, the computation costs and running times of these four strategies are actually similar. For the models in Table(d), the FLOPs of multi-stage fusion are as follows.
>
> |Design | GFLOPs |
> |:-------------------|:-------: |
> | Sequential | 14.988 |
> | Parallel      | 14.988 |
> |Hierarchical KV | 14.996 |
> |Hierarchical Q | 15.163 |

---

> > ### Comment · Reviewer_AXo7 · 2022-12-06
> > **Response to rebuttal**
> >
> > I've checked the rebuttal and it address most of my concerns. I'll keep my initial rating and suggest to accept the paper.

---

### Decision · Program_Chairs · 2023-01-20

**Decision:**

Reject

**Justification For Why Not Higher Score:**

Limited technical novelty and insufficient experimental results.

**Justification For Why Not Lower Score:**

N/A

**Metareview: Summary, Strengths And Weaknesses:**

Initially, the reviews were split between two reviewers who accept ratings (score 8) and the other two who gave weak accept and reject ratings (score 6 and 3). The two reviewers who were favorable (`GtFh` and `AXo7`) commended strong empirical performance of this work on a wide variety of video benchmarks and an extensive ablation study. However, the other two reviewers (`ikef` and `DTkU`) pointed out the limited technical novelty of this work and, to the contrary to the other two reviewers, raised concerns about insufficient experimental results to fully demonstrate the benefit of the modifications introduced over the original Uniformer.

Reviewers were generally appreciative of the strong empirical performance of the proposed approach on public video benchmarks. They pointed out that once the pretrained model checkpoints get released, they will be a valuable contribution to the community.

However, reviewer `ikef` pointed out that the novelty of the proposed local and global uni-blocks is rather limited, and the paper's framing of them could be improved to clarify what the main differences are compared to existing techniques (see the review below for the mentioned earlier work). There was an extensive discussion thread between the reviewer and the authors, and unfortunately the authors' arguments weren't successful to address the reviewer's concerns. In my opinion, this is a valid concern and the paper could benefit from a clear description of technical differences to the mentioned prior work. A similar concern was also shared by the reviewer `DTkU` who pointed out that the technical novelty over the original Uniformer is limited.

There was another major concern (by reviewers `ikef` and `DTkU`) about the insufficient experimental results to fully validate the advantage of the proposed local/global blocks over the original design in Uniformer. Furthermore, reviewer `ikef` raised a valid (and critical!) concern that the most significant gain is achieved when CLIP is used; otherwise the proposed approach even underperforms the original Uniformer with the same compute budget in some benchmarks such as Something-Something-V2 (**meta-reviewer's comment: although this was not brought up during the review period, it's worth mentioning that many prior work have reported that SSv2 requires stronger temporal representations than K400; the weak performance of the proposed approach on SSv2 than on K400 might suggest a potential inefficacy in temporal modeling -- it would be nice if this point can be validated in the future version).

During the discussion period, both the reviewer `AXo7` and `GtFh` argued that the technical aspect of this work is still acceptable (while the individual components aren't novel, they are carefully put together and validated with thorough ablations), but agreed with the concerns about insufficient experiments, especially on the lack of apples-to-apples comparisons (e.g., ST-Adapter, X-CLIP); as a result, they both decreased the rating from 8 to 6.

Overall, this paper provides strong empirical performance on public benchmarks, and there's no doubt that the pretrained checkpoints, once released, will be valuable to the community. However, once we dive deep into the paper, two critical issues arise: 1) limited novelty and 2) insufficient experimental results. For the novelty issue, I feel that it's OK to have some similarities to prior work but the fact that this paper frames the proposed attention modules as something entirely new is problematic and incorrect. This could be fixed with re-writing, but that will require another round of review to evaluate whether technical claims adequately framed and differences to prior work are properly discussed. For the insufficient experiments issue, this meta-reviewer agrees with the reviewers `ikef` and `DTkU` in that, while the results are impressive at a first glance, the paper fails to provide strong evidence showing the advantage of the proposed modifications over Uniformer, which the paper claims as the main contribution.

It was a difficult decision to make -- given the two critical issues, we regret to recommend rejection at this time.




**Summary Of Ac-Reviewer Meeting:**

The discussion was focused on the significance of the technical novelty and insufficient experiments. One reviewer (`AXo7`) argued that the technical aspect of this work is still acceptable (while the individual components aren't novel, they are carefully put together and validated with thorough ablations), but agreed with the concerns about insufficient experiments; as a result, the reviewer decreased the rating from 8 to 6.agreed with the concerns and, as a result, decreased the rating from 8 to 6.